# Inventory of glaciers and perennial snowfields of the conterminous USA

Andrew G. Fountain[1], Bryce Glenn[1], Christopher Mcneil[2]

[1]Department of Geology, Portland State University, 1721 SW Broadway, Portland, OR. 97212 USA, [2]U.S. Geological Survey Alaska Science Center, Anchorage, AK, USA

*Correspondence to:* Andrew G. Fountain (andrew@pdx.edu)

**Abstract**

This report summarizes an updated inventory of glaciers and perennial snowfields of the conterminous United States. The inventory is based on interpretation of mostly aerial imagery provided by the National Agricultural Imagery Program, U.S. Department of Agriculture with some satellite imagery in places where aerial imagery was not suitable. The inventory includes all perennial snow and ice features $\geq 0.01$ km$^2$. Due to aerial survey schedules and seasonal snow cover, imagery acquired over a number of years were required. The earliest date is 2013 and the latest is 2020, but more than 73% of the outlines were acquired from 2015 imagery. The inventory is compiled as shapefiles within a geographic information system that includes feature classification, area, and location. The inventory identified 1331 (366.52 $\pm$ 14.34 km$^2$) glaciers, 1776 (31.01 $\pm$ 9.30 km$^2$) perennial snowfields, and 35 (3.57 km$^2$ $\pm$ no uncertainty) buried-ice features. The data including both the shapefiles and tabulated results are publicly available at https://doi.org/10.15760/geology-data.03 (Fountain & Glenn, 2022).

## 1. Introduction

Glaciers are an important feature of the landscape for several reasons. Geologically, they modify the landscape through erosion and deposition (Alley et al., 2019; Benn & Evans, 2010). Although these processes are typically slow, sudden episodes can occur such as moraine failure due to fluvial erosion resulting in catastrophic debris flows (Beason et al., 2018; Chiarle et al., 2007; O'Connor et al., 2001). Hydrologically, glaciers can be viewed as frozen reservoirs of water that naturally regulate streamflow on seasonal to decadal time scales (Dussaillant et al., 2019; Fountain & Tangborn, 1985; Moore et al., 2009). Glacier runoff increases during warm periods and diminishes during cool, wet periods. Thus, glacier populated watersheds have less seasonally variable runoff than ice-free watersheds. Also, glacier runoff cools stream temperatures in the driest and hottest part of the summer after seasonal snowpacks have vanished (Cadbury et al., 2008; Fellman et al., 2014). As glaciers shrink, they have less ability to buffer seasonal runoff variations and watersheds become more susceptible to drought (Huss & Hock, 2018; Pritchard, 2019). Globally, the loss of perennial ice from the landscape is a major contributor to sea level rise (Meier, 1984; Parkes & Marzeion, 2018; Zemp et al., 2019).

Glacier inventories have been valuable for assessing glacier contribution to sea level change (Hock et al., 2009; Pfeffer et al., 2014), and for assessing regional hydrology (Moore et al., 2009; Yao et al., 2007). They also provide a baseline for quantifying future glacier changes. Updated glacier inventories have been compiled for many regions of the world (Andreassen et al., 2022; Bolch et al., 2010; Smiraglia et al., 2015; Sun et al., 2018). An exception has been western

United States (US), defined here as those conterminous states west of the 100[th] meridian. The most recent inventory is (Fountain et al., 2007, 2017) based on U.S. Geological Survey maps compiled over a 40-year period from the late 1940s to the 1980s. Despite a vigorous history of glacier studies (e.g. Armstrong, 1989; Rasmussen, 2009)), glacial geology (e.g. Bowerman & Clark, 2011; Davis, 1988; Osborn et al., 2012)), and regional inventories (e.g. DeVisser & Fountain, 2015; Fagre et al., 2017; Post et al., 1971) the glacier cover for the entire western US has not been reevaluated.

The earliest scientific identification of glacier-populated regions in the western US date to King (1871) and, more comprehensively, to Russell (1898). The first summary of glacier-covered area for each state was Meier (1961). However, the data sources and methods used to compile the inventories are unknown. Denton (1975) summarized all known glacier studies in the western US but did not tabulate glacier area. Krimmel (2002) updated Meier's study and provided total glacier area for the various mountain ranges by summarizing a variety previous studies published over a 10+ year time span. It is not clear whether the inventory is complete and no data on individual glaciers are provided. Fountain et al. (2007, 2017) compiled the first comprehensive inventory of glaciers in the western US. The data were derived from historical U.S. Geological Survey (USGS) 1:24,000 scale maps compiled over a 40-year period from the 1940s to the 1980s (Gesch et al., 2002; Usery et al., 2009). Because the USGS mapping was based on one-time aerial imagery, the misinterpretation of seasonal snow as perennial was extensive in some regions. The most current study, Selkowitz & Forster (2016), used Landsat satellite imagery compiled over a four-year period, 2010-2014, and an automated detection scheme to define perennial snow and ice. However, these early automated schemes are known to misclassify debris-covered ice as ice-free landscape underestimating glacier area (Earl & Gardner, 2016; Paul et al., 2007; Rabatel et al., 2017). Recent advances in automated detection have reduced these errors suggesting a more promising future (Lu et al., 2022; Robson et al., 2020).

This paper presents the results of an updated and comprehensive inventory of glaciers and perennial snowfields of the western US for the purpose of defining their current extent and to provide of baseline for estimating future changes. We summarize our methods, uncertainties, tabulated results, and data availability. The data referenced throughout the manuscript are publicly available at https://doi.org/10.15760/geology-data.03.

## 2. Methods

2.1 Data Sources, Classification, Digitizing, and Completeness

The glaciers and perennial snowfields were initially located using a geographic information system (GIS) database from Fountain et al. (2007, 2017). New outlines were manually digitized from three sources of optical imagery. Most of the outlines were digitized from color digital orthographic aerial photographs available from the National Agricultural Imagery Program (NAIP), U.S. Department of Agriculture, Farm Service Agency program (NAIP, 2017), (https://datagateway.nrcs.usda.gov/GDGHome_DirectDownLoad.aspx). Since 2009, the imagery is collected on cycles of two to three years. The aerial imagery was orthorectified using the inertial navigation system - GPS unit in the aircraft. Photo identifiable GPS-survey ground control points were then used to adjust the photo strip. Orthorectified strips, which had $\geq 30\%$

overlap with adjacent strips, were overlaid with each other and with ground control points to
check accuracy. The image strips are then mosaicked together. The spatial resolution was ≤ 0.6
84  m with a horizontal accuracy of ≤ 6 m of photo-identifiable ground control points (NAIP, 2017).
The NAIP imagery fit the historic USGS glacier outlines remarkably well. In a few cases, NAIP
imagery was not suitable due to seasonal snow, deep shadows, or image warping caused by
orthophoto rectification, therefore other sources were used including Maxar satellite imagery
(Maxar Technologies, Inc) with a spatial resolution of 0.5 – 1 m. For 21 perennial snowfields
and three glaciers we relied on the most recent snow-free imagery available in Google Earth
(Google, Inc), resolution ~ 1m, because no other imagery was suitable. The outlines were
digitized in Google Earth and exported to ArcMap (Esri, Inc).

We manually identified all glaciers, ice patches, and perennial snowfields. Glaciers are defined
as perennial snow and ice that moves (Cogley et al., 2011). A feature was considered perennial if
it was present on the original 1:24,000 USGS topographic maps and present on all Google Earth
imagery. Movement was identified by the presence of crevasses. Perennial snowfields and ice
patches do not exhibit movement, as indicated by a lack of crevasses observed in the imagery.
We do not distinguish between snowfields and ice patches and refer to both as perennial
snowfields.

Contiguous glacier cover, most commonly on volcanoes, was separated into individual glaciers if
they had unique names as indicated on the USGS maps. The orientation of crevasse patterns was
used to define flow divides. In the absence of these patterns, shaded relief maps from digital
elevation models were used. These models were derived from aerial lidar data, flown under
contract to the USGS (Bard, 2017a, 2017b, 2019; Robinson, 2014) or the Oregon Department of
Geology and Mineral Industries (DOGAMI, 2011).

We encountered a number of challenges to our classification and delineation of the glaciers and
perennial snowfields. Although crevasses were used to define movement, in a few cases it
appeared that they penetrated through the feature to the bedrock underneath suggesting a
mechanical break up. In these cases, the feature was classified as a snowfield.  Debris-cover
made defining the glacier outline for some glaciers on the volcanoes of the Cascade Range. We
relied on local knowledge to help define some boundaries and independent digitization efforts by
the authors and others to provide an uncertainty as explained below. In the high alpine regions of
California, Colorado, and Wyoming, the terminus of some glaciers was hard to define. Rather
than abruptly terminating, the ice seems to thin and smoothly transitions into the surrounding
rock talus (Figure 1). It was unclear whether a thin debris layer blanketed the ice or cobbles and
boulders protruded through the thin ice. The boundary was mapped along the edge of identifiable
ice.

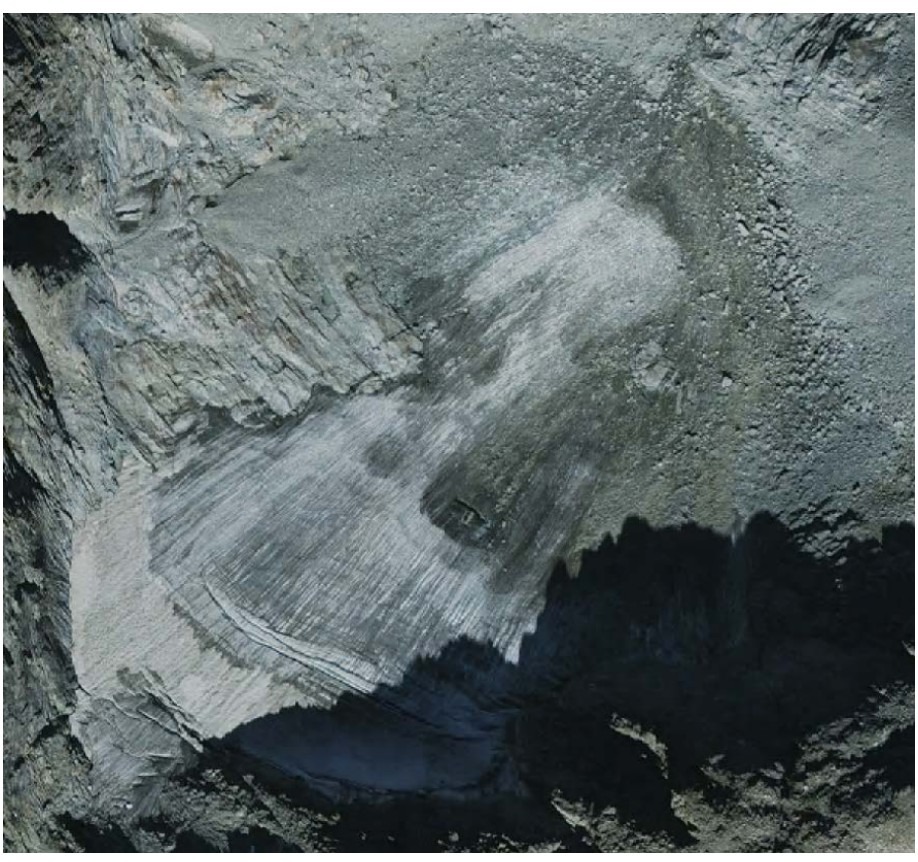

**Figure 1.** An example of a glacier seemingly melting into the talus surrounding the terminus
(upper right).  The glacier is flowing from the lower left-hand corner to the upper right-hand
corner. The glacier is located in the Wind River Range, WY, INV_ID E618081N4774579 and
base image is from the National Agricultural Image Program taken in 2015.

In a few situations, we found it difficult to distinguish glaciers from rock glaciers (Brardinoni et
al., 2019). A rock glacier is a mass of rock debris in a matrix of ice that flows (Cogley et al.,
2011). They can be difficult to distinguish from a debris-covered glacier, one that has extensive
rock debris over the ablation zone, that lower part of a glacier with exposed ice in late summer.
We adopted the following topographic classification. If the slope of the apparent ice
patch/snowfield was similar to the slope of the rock glacier then we considered it part of the rock
glacier (Figure 2a). On the other hand, if a topographic depression separates the apparent
glacier/snowfield from the start of a rock glacier, then it was considered independent feature
(Figure 2b). This latter case is similar to the "glacier forefield-connected" rock glacier as
described by (RGIK, 2022).

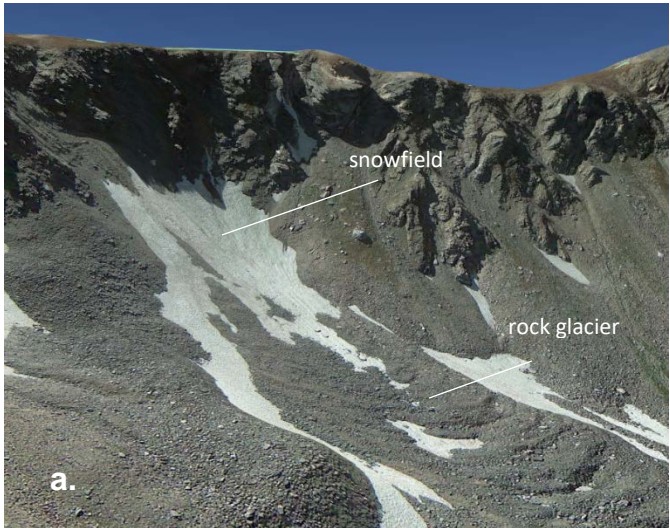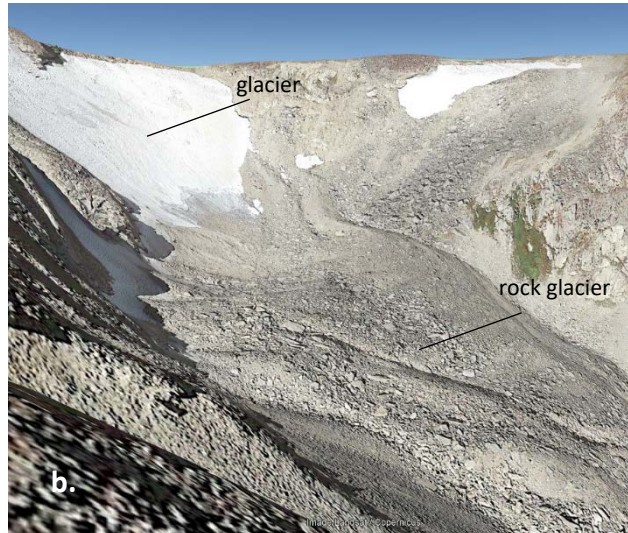

**Figure 2.** Examples of glacier versus rock glacier identification. (a) An example of a snowfield that is considered part of the rock glacier. Location, Colorado Front Range, 40.827477° N, -106.657400° E. Image is from © Google Earth, 9/2014; (b) Tyndall Glacier in the Colorado Front Range, 40.305291° N, -105.689602° E, with a rock glacier slightly down valley. Image is from © Google Earth 9/2016.

In a number of situations, we observed buried ice adjacent to a glacier (Figure 3). Here we use the term 'buried ice' to mean dead ice formerly part of a flowing glacier, and not the permafrost context of ice embedded within or on top of perennially frozen ground. The rocky surface texture of the buried ice was hummocky and very different from surrounding bedrock and adjacent ice, and not a moraine. Occasionally a crack in the surface revealed subsurface ice. The feature appeared to be non-moving (dead) ice that is covered by debris similar to some of the ice-debris complexes described by Bolch et al. (2019). We decided to include these features as a separate classification, 'buried ice', because their size was large relative to the glacier, they were probably once part of the glacier, and may be important local sources of meltwater for streamflow.

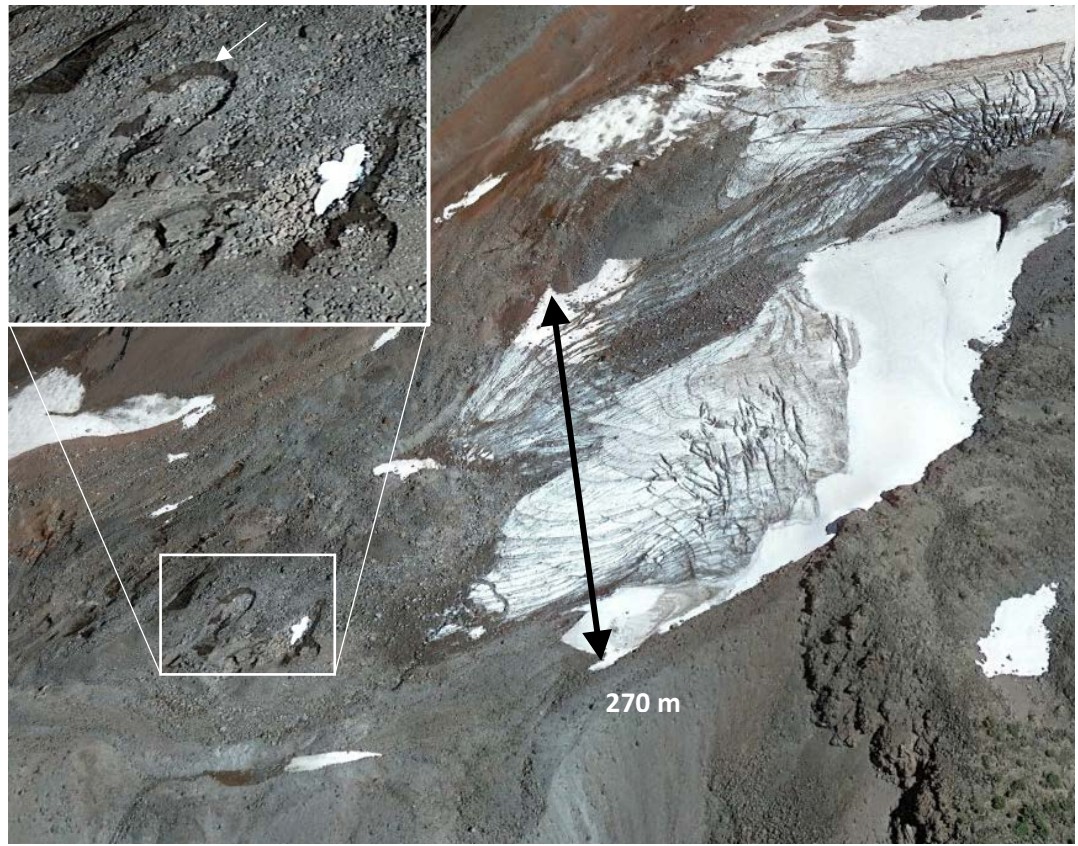

**Figure 3.** Lost Creek Glacier, South Sister, Oregon. Note buried ice and lack of crevasses to the left of the grey-blue ice, suggesting ice that is no longer moving and therefore not part of the dynamic glacier. The white box surrounds an area that has collapsed due to subsurface melt. The inset enlargement shows a cliff edge of exposed dirty ice (white arrow in upper left) indicated by a darker color suggesting wet sediment and a finer texture than the surface debris. The black arrow shows the width of the cleaner ice for scale.  Image is from © Google Earth, 8/9/2021.

The glaciers and perennial snowfields outlines were digitized using ArcMap (Esri, Inc), a geographic information system, at scales varying from 1:300 to 1:2000 depending on image quality and complexity. We used the native projection of the image, North American Datum of 1983 (NAD83) for NAIP, and World Geodetic System 1984 (WGS84) for Maxar and Google Earth. When Maxar or Google Earth imagery were used, final outlines were projected into the NAD83 coordinate system. Google Earth was often used an additional aid in interpretation because of its tilt and rotation features yielded oblique perspectives. Retaining only those outlines ≥ 0.01 km$^2$, each was checked independently by the two senior authors of this report and in some cases by a third collaborator in order to reduce bias (Leigh et al., 2019). If an outline was revised, then it was returned to its original author for review and correction, and the process iterated until all parties agreed.

Our initial inventory was then compared sequentially to two other independent inventories to test for errors of omission or commission. The first comparison was to the Selkowitz and Forster (2016) inventory (SFI). However, to compare the inventories we had to first reconcile the

differences in methods. Buried-ice features were eliminated from our inventory because the SFI did not map buried ice. The SFI was filtered to only include features ≥ 0.01 km$^2$ to match our minimum area threshold; a small number of features located in Canada were removed; and a few mis-classifications of ponds, lakes and dry lakebeds as glaciers were removed. Notably, the SFI did not split contiguous ice masses, such as glacier-covered volcanoes, into individual glaciers, consequently we do not expect the number of features in the SFI and our inventory to match. Once the two inventories were reconciled, those glaciers and perennial snowfields unique to one inventory were examined for inclusion in a revised inventory. Features selected from the SFI were digitized using the same imagery we used for our inventory.

The revised inventory was then compared to the 2016 National Land Cover Database (NLCD, Dewitz, 2019), which did not map glaciers and perennial snowfields per se, but mapped the distribution of perennial snow and ice (Jin et al., 2019; Wickham et al., 2021). However, the NLCD used a small number of recent images to assess a 'perennial' presence and therefore significant errors of commission are expected. Also, the landscape class of snow and ice received less attention than other classes (e.g. agriculture) such that the timing of imagery acquisition may be earlier in the summer than optimal and misclassification of clouds as snow and ice may be present (personal communication C. Homer and J. Dewitz, USGS, email December 2015). The NLCD inventory was compared to the revised inventory and, as before, the features unique to one inventory were examined for inclusion. Those features selected from the NLCD for inclusion were digitized using the same imagery we used for our inventory.

2.2 Uncertainty

Three main sources of uncertainty in the glacier outlines, are georeferencing, digitization, and interpretation (DeVisser & Fountain, 2015; Sitts et al., 2010). We found georeferencing error to very small. In any case, the precise location of the outline does not affect its area. Also, the digitized points are highly correlated such that no deviations from the true outline are caused by georeferencing. Digitizing error is relatively small, 1%, with good imagery and crisp contrast between the glacier and ice-free surroundings (DeVisser & Fountain, 2015; Hoffman et al., 2007). The largest uncertainty is interpretation error caused by poor imagery, shadow, debris cover, and seasonal snow patches. This uncertainty was calculated in different ways according to the situation. If the outline was digitized a second (or third) time due different interpretations by the authors or collaborators the uncertainty is one-half the absolute difference of the between the largest and smallest digitized areas (the range) divided by the final area and expressed as a percentage. For the relatively few glaciers where a small section of perimeter was masked by deep shadow, seasonal snow patches, rock debris, or poor imagery, a higher uncertainty was assigned by visually estimating the area in question and dividing by the total possible area. In a few cases the location of a flow divide between glaciers wasn't clear a 5% error is assigned. This was calculated from the area difference in several test cases where multiple possible flow divides were digitized. For perennial snowfields, the smaller patch of perennial snow is often covered by seasonal snow, which varies greatly from year to year. We measured the area of a number of snowfields over time using late summer historic imagery in Google Earth. Results showed that the variations in snowfield area could be as much as 30%. We assigned this somewhat arbitrary uncertainty in order to note snowfield presence and location, but preclude them from area change calculations because area differences are typically smaller than the assigned uncertainty.

## 3. Results

Our initial inventory identified 2267 glaciers and perennial snowfields totaling 391.95 km$^2$.
About 70% (1576) overlapped the features in the SFI. After examining all features unique to
each inventory, we revised our inventory to include 2373 (394.99 km$^2$) glaciers and perennial
snowfields. Comparing the revised inventory to the 2016 NLCD resulted in adding another 134
(2.53 km$^2$) features, which included 12 (0.38 km$^2$) glaciers. The final inventory includes 2542
features composed of 1331 (366.52 km$^2$) glaciers, 1176 (31.01 km$^2$) perennial snowfields, and 35
(3.57 km$^2$) buried ice deposits (Table 1; Figure 4). Most glaciers and perennial snowfields, 1554
(62%) were outlined using 2015 NAIP imagery with the remainder outlined using mostly NAIP
imagery from 2013 to 2020.

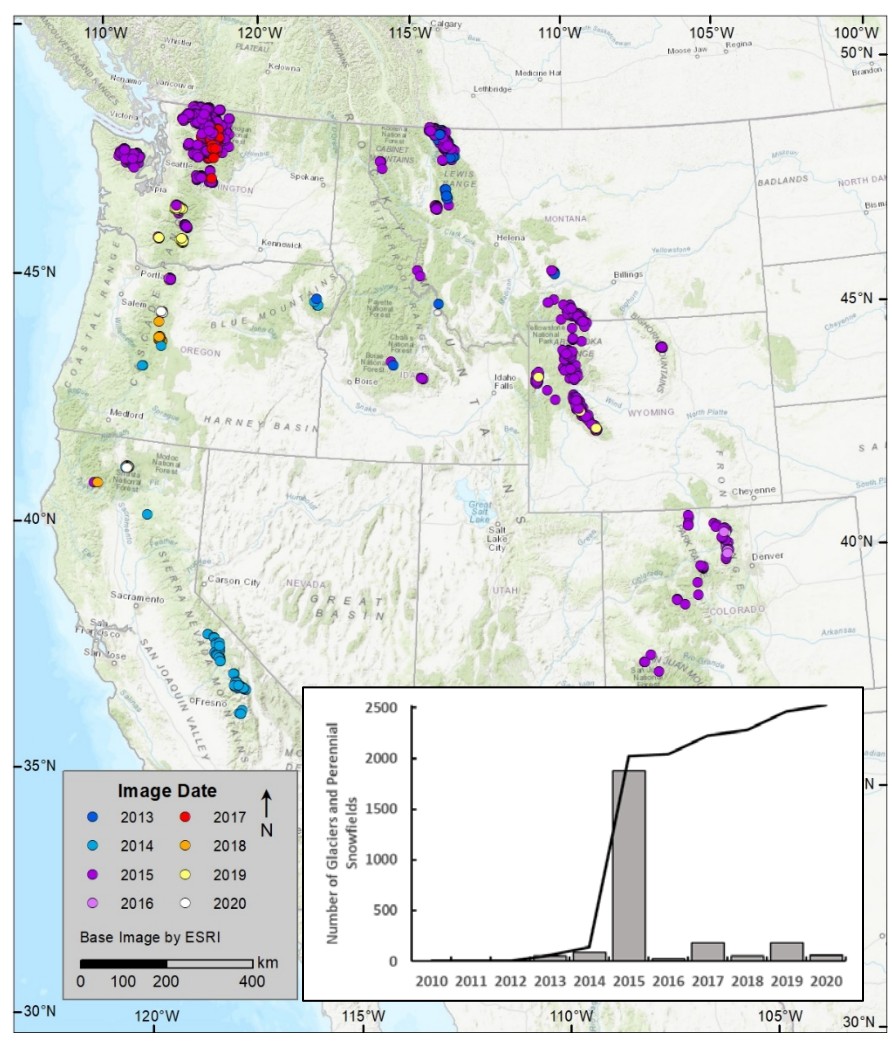

**Figure 4**. The spatial distribution and number of glaciers and perennial snowfields, greater than 0.01 km$^2$, in the western United States. Colors indicate the date of aerial and satellite imagery used to outline the features. The line is the cumulative total. Base imagery from Esri Inc.  Inset is a bar graph and cumulative sum of the number of glaciers and perennial snowfields digitized in each image date.

**Table 1.** The summary of the glacier inventory for the American West, exclusive of Alaska. Number is the total number of features within each classification (Class), 'Max Area' is the largest area of the feature within that class and 'Mean Area' is the average area. Note that the uncertainty of 'Buried ice' is unknown.

| State/Region/Class | Number | Total Area km$^2$ | Max Area km$^2$ | Mean Area km$^2$ |
|---|---|---|---|---|
| **California** | **132** | **10.63 ± 0.61** | **1.45** | **0.08** |
| **Cascade Range** | **39** | **5.74 ± 0.37** | **1.45** | **0.15** |
| Buried ice | 5 | 0.44 | 0.16 | 0.09 |
| Glaciers | 10 | 4.61 ± 0.17 | 1.45 | 0.46 |
| Perennial snowfields | 24 | 0.68 ± 0.21 | 0.08 | 0.03 |
| **Sierra Nevada** | **91** | **4.86 ± 0.23** | **0.66** | **0.05** |
| Buried ice | 2 | 0.13 | 0.10 | 0.06 |
| Glaciers | 64 | 4.37 ± 0.12 | 0.66 | 0.07 |
| Perennial snowfields | 25 | 0.37 ± 0.11 | 0.03 | 0.01 |
| **Trinity Alps** | **2** | **0.03 ± 0.00** | **0.02** | **0.02** |
| Glaciers | 2 | 0.03 ± 0.00 | 0.02 | 0.02 |
| **Colorado** | **84** | **2.20 ± 0.46** | **0.16** | **0.03** |
| **Elk Mountains** | **5** | **0.09 ± 0.03** | **0.03** | **0.02** |
| Glaciers | 1 | 0.01 ± 0.00 | 0.01 | 0.01 |
| Perennial snowfields | 4 | 0.08 ± 0.02 | 0.03 | 0.02 |
| **Front Range** | **58** | **1.73 ± 0.33** | **0.16** | **0.03** |
| Glaciers | 13 | 0.74 ± 0.03 | 0.16 | 0.06 |
| Perennial snowfields | 45 | 0.99 ± 0.30 | 0.09 | 0.02 |
| **Gore Range** | **7** | **0.11 ± 0.03** | **0.02** | **0.02** |
| Glaciers | 1 | 0.02 ± 0.00 | 0.02 | 0.02 |
| Perennial snowfields | 6 | 0.09 ± 0.03 | 0.02 | 0.02 |
| **Medicine Bow Mountains** | **1** | **0.04 ± 0.01** | **0.04** | **0.04** |
| Perennial snowfields | 1 | 0.04 ± 0.01 | 0.04 | 0.04 |
| **Park Range** | **6** | **0.11 ± 0.03** | **0.03** | **0.02** |
| Perennial snowfields | 6 | 0.11 ± 0.03 | 0.03 | 0.02 |
| **San Miguel Mountains** | **5** | **0.07 ± 0.02** | **0.02** | **0.01** |
| Perennial snowfields | 5 | 0.07 ± 0.02 | 0.02 | 0.01 |
| **Sawatch Range** | **2** | **0.04 ± 0.01** | **0.03** | **0.02** |
| Perennial snowfields | 2 | 0.04 ± 0.01 | 0.03 | 0.02 |
| **Idaho** | **6** | **0.08 ± 0.02** | **0.02** | **0.01** |
| **Sawtooth Range** | **6** | **0.08 ± 0.02** | **0.02** | **0.01** |

| | | | | |
|---|---|---|---|---|
| Perennial snowfields | 6 | 0.08 ± 0.02 | 0.02 | 0.01 |
| **Montana** | **416** | **30.26 ± 2.27** | **1.45** | **0.07** |
| **Beartooth -Absaroka** | **111** | **6.07 ± 0.64** | **0.45** | **0.05** |
| Buried ice | 1 | 0.04 | 0.04 | 0.04 |
| Glaciers | 50 | 4.31 ± 0.12 | 0.45 | 0.09 |
| Perennial snowfields | 60 | 1.72 ± 0.52 | 0.22 | 0.03 |
| **Bitterroot Range** | **4** | **0.08 ± 0.02** | **0.03** | **0.02** |
| Glaciers | 1 | 0.03 ± 0.00 | 0.03 | 0.03 |
| Perennial snowfields | 3 | 0.05 ± 0.02 | 0.02 | 0.02 |
| **Cabinet Mountains** | **9** | **0.25 ± 0.08** | **0.08** | **0.03** |
| Perennial snowfields | 9 | 0.25 ± 0.08 | 0.08 | 0.03 |
| **Crazy Mountains** | **13** | **0.27 ± 0.06** | **0.04** | **0.02** |
| Glaciers | 3 | 0.06 ± 0.00 | 0.04 | 0.02 |
| Perennial snowfields | 10 | 0.21 ± 0.06 | 0.04 | 0.02 |
| **Lewis Range** | **230** | **21.38 ± 1.15** | **1.45** | **0.09** |
| Glaciers | 145 | 19.22 ± 0.50 | 1.45 | 0.13 |
| Perennial snowfields | 85 | 2.16 ± 0.65 | 0.09 | 0.03 |
| **Mission-Swan-Flathead** | **49** | **2.20 ± 0.34** | **0.22** | **0.04** |
| Glaciers | 11 | 1.16 ± 0.02 | 0.22 | 0.11 |
| Perennial snowfields | 38 | 1.04 ± 0.31 | 0.09 | 0.03 |
| **Oregon** | **116** | **15.38 ± 1.62** | **1.16** | **0.13** |
| **Cascade Range** | **110** | **15.24 ± 1.58** | **1.16** | **0.14** |
| Buried ice | 7 | 1.25 | 0.45 | 0.18 |
| Glaciers | 42 | 11.90 ± 0.95 | 1.16 | 0.28 |
| Perennial snowfields | 61 | 2.09 ± 0.63 | 0.15 | 0.03 |
| **Wallowa Mountains** | **6** | **0.14 ± 0.63** | **0.04** | **0.02** |
| Perennial snowfields | 6 | 0.14 ± 0.04 | 0.04 | 0.02 |
| **Washington** | **1481** | **312.26 ± 16.33** | **11.24** | **0.21** |
| **Cascade Range-Northern** | **1126** | **186.58 ± 9.64** | **6.06** | **0.17** |
| Buried ice | 10 | 0.50 | 0.15 | 0.05 |
| Glaciers | 706 | 176.27 ± 6.70 | 6.06 | 0.25 |
| Perennial snowfields | 410 | 9.80 ± 2.94 | 0.16 | 0.02 |
| **Cascade Range-Southern** | **219** | **101.66 ± 5.86** | **11.24** | **0.46** |
| Buried ice | 10 | 1.20 | 0.30 | 0.12 |
| Glaciers | 69 | 95.64 ± 4.42 | 11.24 | 1.39 |
| Perennial snowfields | 140 | 4.82 ± 1.45 | 0.33 | 0.03 |
| **Olympic Mountains** | **136** | **24.02 ± 0.82** | **5.09** | **0.18** |
| Glacier | 106 | 23.44 ± 0.65 | 5.09 | 0.22 |
| Perennial snowfield | 30 | 0.57 ± 0.17 | 0.06 | 0.02 |
| **Wyoming** | **307** | **30.29 ± 2.34** | **2.32** | **0.10** |
| **Absaroka Range** | **62** | **1.44 ± 0.33** | **0.12** | **0.02** |
| Glacier | 10 | 0.48 ± 0.05 | 0.12 | 0.05 |
| Perennial snowfield | 52 | 0.96 ± 0.29 | 0.05 | 0.02 |
| **Bighorn Mountains** | **8** | **0.42 ± 0.03** | **0.22** | **0.05** |
| Glacier | 3 | 0.34 ± 0.01 | 0.22 | 0.11 |

| | | | | |
|---|---|---|---|---|
| Perennial snowfield | 5 | 0.08 ± 0.02 | 0.03 | 0.02 |
| **Teton Range** | **49** | **2.04 ± 0.21** | **0.23** | **0.04** |
| Glacier | 20 | 1.46 ± 0.03 | 0.23 | 0.07 |
| Perennial snowfield | 29 | 0.59 ± 0.18 | 0.05 | 0.02 |
| **Wind River Range** | **188** | **26.39 ± 1.76** | **2.32** | **0.14** |
| Glacier | 74 | 22.42 ± 0.57 | 2.32 | 0.30 |
| Perennial snowfield | 114 | 3.97 ± 1.19 | 0.26 | 0.03 |
| **Grand Total** | **2542** | **401.10 ± 23.64** | **11.24** | **0.16** |

Before summarizing the inventory data, a note about the content in Appendix A. It summarizes
the officially named glaciers that we regard as snowfields or missing; labeling issues found in the
USGS Geographic Names Information System, the official agency responsible for hosting the
names and locations of landscape features; and detailed notes, organized by US State, on the
specific imagery used and challenges encountered digitizing glacier and snowfield outlines.

The glaciers and perennial snowfields are generally small, averaging 0.28 and 0.03 $km^2$,
respectively. Like glaciers elsewhere in the northern hemisphere, most glaciers face north to east.
(Evans, 2006; Fountain et al., 2017; Schiefer et al., 2007).  The distribution of glacier area is
skewed toward smaller ice masses (Figure 5a). The State of Washington in the Pacific Northwest
has the largest number of glaciers, ice area and the largest glacier (11.24 $km^2$ Emmons Glacier)
of any of the other states (Table 1). Indeed, the glacier cover on Mount Rainier alone (77.37 $km^2$)
is greater than the total sum in all the other states (71.16 $km^2$). The elevation distribution of
glacier-covered area is bimodal with maxima at 2400 m and 3650 m (Figure 5b). The spatial
distribution of elevations shows a regional climate control with the lowest glaciers and perennial
snowfields in the maritime climate of the Pacific Northwest of Washington, Oregon, northern
California, and western Montana and the high elevations located in the continental climate of
central California, Colorado, Wyoming and southern Montana (Figure 6).

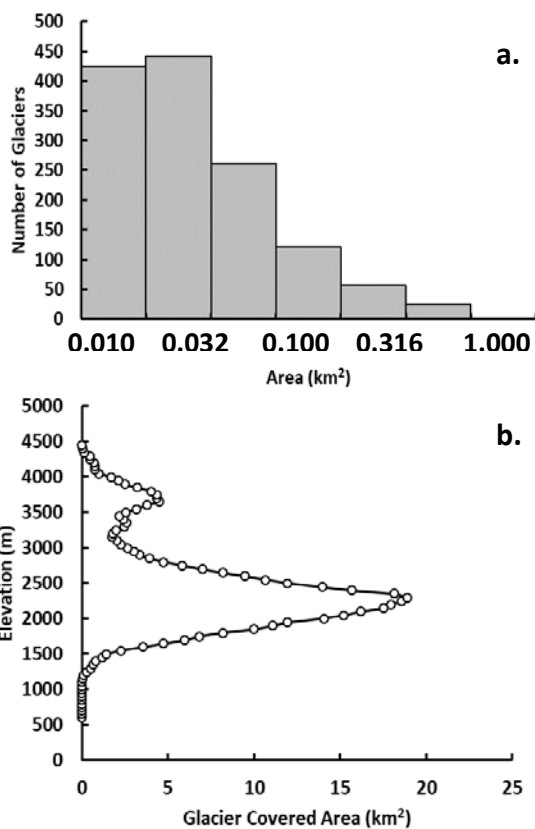

Figure 5. The area and elevation distribution of glaciers in the western U.S., (a) Histogram
showing the number of glaciers as a function of area. The x-axis intervals are log intervals; (b)
Elevation distribution of glacier-covered area.

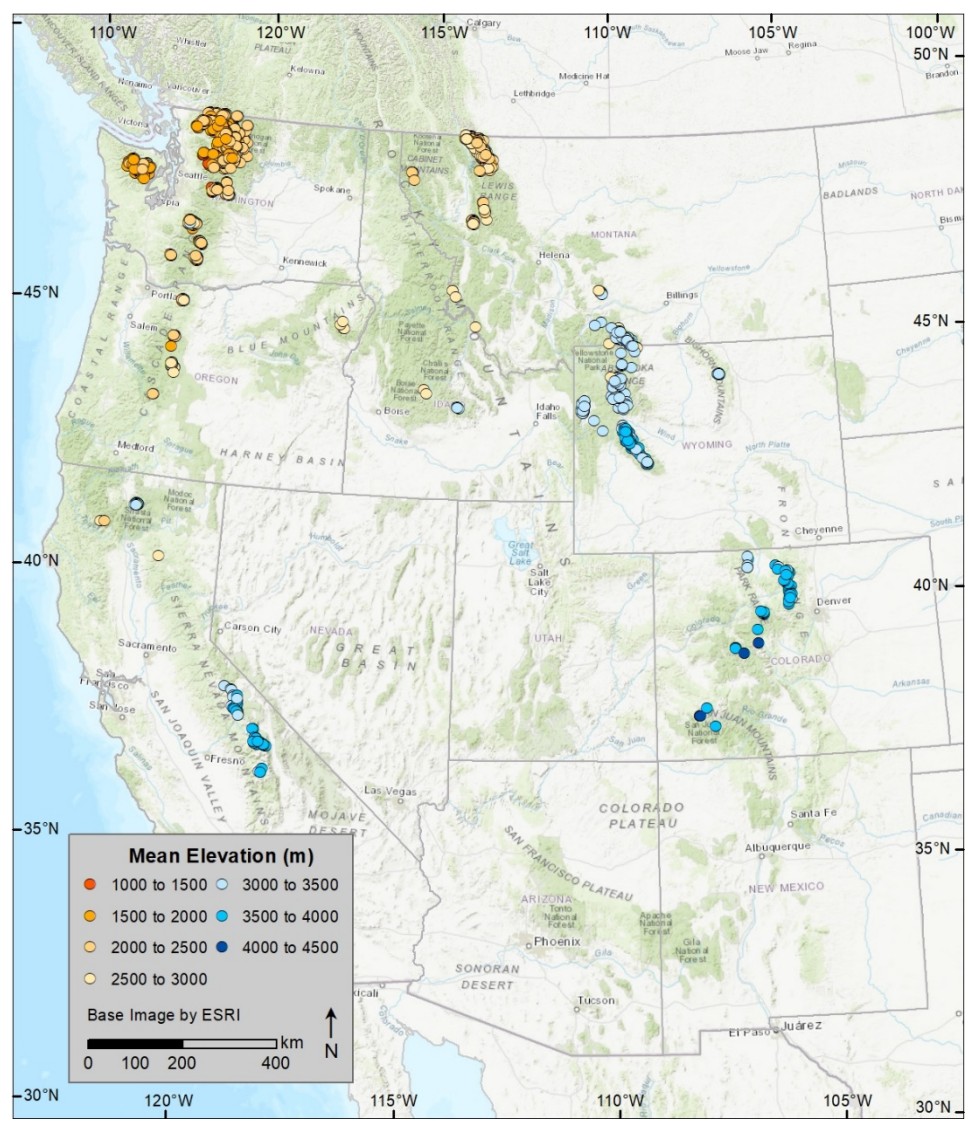

Figure 6. Elevation distribution of glaciers and perennial snowfields across the western US. Base imagery from Esri Inc.

The final inventory conflicts with the current database of the Geographic Names Information System (https://www.usgs.gov/us-board-on-geographic-names/domestic-names). The inventory excludes 52 officially named glaciers because 2 have disappeared, 25 were classified as perennial snowfields, the area of 18 was less than 0.01 km$^2$, and 7 were considered rock glaciers (Appendix A,Table A1). In some cases, a named glacier or snowfield had split into multiple pieces since the original USGS mapping; all pieces were assigned the same name in the inventory (Appendix A, Table A2). Several labels that identify the name of the glacier are not clearly associated with a specific glacier and these are listed in Table 7.3.

**4.  Discussion**
The advent of relatively frequent high resolution ($\leq 1$ m) optical aerial and satellite imagery
available at little or no cost has made compiling and updating glacier inventories a realistic
opportunity. Finding suitable imagery spanning only a few years apart provides a near-snapshot
of glacier cover. This contrasts strongly with mapping efforts only a few decades ago when
aerial-only photographic surveys required decades to cover the western US (Gesch et al., 2002).
And the advent of GIS software made digitizing, summarizing, and interrogating digital outlines
practical.
We had used the Fountain et al., 2017 historic inventory as a template to locate and update the
perimeters of all the glaciers and perennial snowfields. Considering that the inventory was
derived from the U.S. Geological Survey 1:24,000 maps, a result of a national effort to remap the
entire country at a higher resolution, we were a surprised that 240 features (~10%) were missed.
These missing features were revealed after comparison with two other independently derived
inventories. We had a similar experience in a prior study when comparing two independently
derived glacier inventories. Together they suggest that independent efforts are important to
compiling a comprehensive inventory.
Multiple checks more accurately define glacier perimeters (Leigh et al., 2019). Different
investigators may make different decisions about glacier boundaries and results can differ
particularly in debris-covered conditions or along flow divides (Paul et al., 2013). When they
agree, it provides some confidence of the interpretation accuracy and where they disagree it
provides input for estimating interpretation error.
The total area of glaciers in the western US, 367 km$^2$, is a little smaller than that in Austria, 415
318 km$^2$, (Fischer et al., 2015). Like glacier populated regions elsewhere the distribution of glacier
area is skewed towards smaller glaciers (e.g. Linsbauer et al., 2012; Mishra et al., 2023; Zalazar
et al., 2020). The uncertainty in glacier area is also similar with an overall 5% uncertainty for the
total area. Paul et al. (2020) report an uncertainty of 3.3% over a set of 15 glaciers, 4% for 7
glaciers (Zalazar et al., 2020), 2.3% for 15 glaciers (Linsbauer et al., 2021). Our assessment
method differs from those cited here in that we estimate the uncertainty for each individual
glacier rather than upscaling the uncertainty calculated for a small subsample.
**5.  Data products and availability**
The data are available in three formats. The geospatial data and attribute tables are available in
the shapefile (Esri) format and in an open source GeoJSON format.  The attribute table is also
available as an EXCEL file. These data products can be obtained from
https://doi.org/10.15760/geology-data.03 (Fountain & Glenn, 2022) and from the Global Land
Ice Measurements from Space website http://glims.colorado.edu/glacierdata/. Maxar imagery
was accessed through the USGS and NGA NEXTVIEW license. The Maxar imagery has limited
availability owing to restrictions (proprietary interest). Contact cmcneil@usgs.gov for more

information.
**6.  Conclusions**
We have compiled a new and comprehensive inventory of glaciers and perennial snowfields in
the western US from aerial and satellite imagery. Results show that 2542 features are currently
present and include 1331 (366.52 km$^2$) glaciers, 1176 (31.01 km$^2$) perennial snowfields, and 35
(3.57 km$^2$) buried ice deposits. Most of the data were acquired from 2015 NAIP imagery with the
remainder from NAIP imagery and a few satellite images acquired over the period of 2013 to
2020. The state of Washington has the greatest number and area of glaciers and perennial
snowfields. This product updates an older inventory based on USGS 1:24000 maps compiled in
the middle-late 1900's. The new inventory is a significant improvement in accuracy because the
archive of historical imagery in Google Earth greatly aided our efforts to classify glaciers versus
perennial snowfields. Finally, this new inventory provides a baseline for assessing glacier change
in the coterminous US.
**7.  Appendix A**
**A1 Missing Glaciers**
**Table A1** List of officially named glaciers not classified as glaciers and excluded from the final
inventory. Names come from the Geographic Names Information System, (US Geological
Survey, 2022). The 'Reason' column lists why the named glacier is no longer considered a glacier
in our inventory.

| State/Region/Glacier Name | Reason |
|---|---|
| **California** | |
| **Sierra Nevada** | |
| Matthes Glaciers | rock glacier |
| Mount Warlow Glacier | rock glacier |
| Powell Glacier | rock glacier |
| **Colorado** | |
| **Front Range** | |
| Isabelle Glacier | perennial snowfield |
| Mills Glacier | perennial snowfield |
| Moomaw Glacier | perennial snowfield |
| Peck Glacier | perennial snowfield |
| Rowe Glacier | < 0.01 km$^2$ |
| Saint Marys Glacier | < 0.01 km$^2$ |
| Taylor Glacier | rock glacier |
| The Dove | < 0.01 km$^2$ |
| **Idaho** | |
| **Lost River Range** | |
| Borah Glacier | rock glacier |
| **Montana** | |

**Beartooth Mountains-Absaroka Range**
    Grasshopper Glacier                         rock glacier
**Cabinet Mountains**
    Blackwell Glacier                          perennial snowfield
**Crazy Mountains**
    Grasshopper Glacier                         rock glacier
**Lewis Range**
    Boulder Glacier                           perennial snowfield
**Mission-Swan-Flathead Ranges**
    Fissure Glacier                           $< 0.01 \text{ km}^2$
    Gray Wolf Glacier                         perennial snowfield

**Oregon**

**Cascade Range**
    Carver Glacier                           perennial snowfield
    Clark Glacier                            perennial snowfield
    Irving Glacier                           perennial snowfield
    Lathrop Glacier                          $< 0.01 \text{ km}^2$
    Palmer Glacier                           perennial snowfield
    Skinner Glacier                          perennial snowfield
    Thayer Glacier                           $< 0.01 \text{ km}^2$
**Wallowa Mountains**
    Benson Glacier                           perennial snowfield

**Washington**

**Cascade Range-Northern**
    Lyall Glacier                            perennial snowfield
    Milk Lake Glacier                         disappeared
    Snow Creek Glacier                      perennial snowfield
    Spider Glacier                           perennial snowfield
    Table Mountain Glacier                  $< 0.01 \text{ km}^2$
**Cascade Range-Southern**
    Ape Glacier                             $< 0.01 \text{ km}^2$
    Dryer Glacier                            perennial snowfield
    Forsyth Glacier                          $< 0.01 \text{ km}^2$
    Meade Glacier                            perennial snowfield
    Nelson Glacier                           $< 0.01 \text{ km}^2$
    Packwood Glacier                       perennial snowfield
    Pinnacle Glacier                        $< 0.01 \text{ km}^2$
    Pyramid Glaciers                        $< 0.01 \text{ km}^2$
    Shoestring Glacier                      $< 0.01 \text{ km}^2$
    Stevens Glacier                          perennial snowfield
    Talus Glacier                            perennial snowfield
    Unicorn Glacier                          $< 0.01 \text{ km}^2$
    Williwakas Glacier                      perennial snowfield
**Olympic Mountains**
    Anderson Glacier                       perennial snowfield

| | |
|---|---|
| Lillian Glacier | $< 0.01$ km$^2$ |
| **Wyoming** | |
| **Absaroka Range** | |
| DuNoir Glacier | $< 0.01$ km$^2$ |
| **Teton Range** | |
| Petersen Glacier | $< 0.01$ km$^2$ |
| Teepe Glacier | perennial snowfield |
| **Wind River Range** | |
| Hooker Glacier | disappeared |
| Harrower Glacier | perennial snowfield |
| Tiny Glacier | $< 0.01$ km$^2$ |

**A2 Glaciers that have split into multiple pieces and current errors in glacier label names**

**Table A2.** List of named glaciers that have split into multiple pieces. Names come from the Geographic Names Information System (https://www.usgs.gov/tools/geographic-names-information-system-gnis). 'Count' refers to the number of pieces in the updated inventory. 'Classes' is the classification of the pieces; glacier, perennial snowfield, buried-ice, or a combination.

| State/Region/Glacier Name | Count | Classes |
|---|---|---|
| **California** | | |
| **Cascade Range** | | |
| Bolam Glacier | 2 | Glaciers and perennial snowfields |
| Hotlum Glacier | 2 | Glaciers and perennial snowfields |
| Whitney Glacier | 2 | Glaciers and perennial snowfields |
| Wintun Glacier | 3 | Glaciers and perennial snowfields |
| **Sierra Nevada** | | |
| Goethe Glacier | 2 | Glaciers only |
| Lyell Glacier | 4 | Glaciers and perennial snowfields |
| Norman Clyde Glacier | 3 | Glaciers only |
| Powell Glacier | 2 | Glacier and Buried-ice |
| **Colorado** | | |
| **Front Range** | | |
| Saint Vrain Glaciers | 6 | Glaciers and perennial snowfields |
| **Montana** | | |
| **Beartooth Mountains-Absaroka Range** | | |
| Castle Rock Glacier | 3 | Glaciers and perennial snowfields |
| Granite Glacier | 2 | Glaciers only |
| Grasshopper Glacier | 4 | Glaciers and perennial snowfields |
| Hopper Glacier | 2 | Glaciers and perennial snowfields |
| Snowbank Glacier | 2 | Glaciers only |
| Wolf Glacier | 2 | Glaciers only |
| **Lewis Range** | | |
| Agassiz Glacier | 3 | Glaciers only |
| Blackfoot Glacier | 2 | Glaciers only |

| | | |
|---|---|---|
| Carter Glaciers | 2 | Glaciers and perennial snowfields |
| Dixon Glacier | 3 | Glaciers and perennial snowfields |
| Harrison Glacier | 5 | Glaciers and perennial snowfields |
| Kintla Glacier | 2 | Glaciers only |
| Logan Glacier | 2 | Glaciers only |
| Shepard Glacier | 3 | Glaciers only |
| Siyeh Glacier | 2 | Glaciers only |
| Two Ocean Glacier | 2 | Glaciers only |
| Whitecrow Glacier | 5 | Glaciers and perennial snowfields |

**Mission Range-Swan Range-Flathead Range**

| | | |
|---|---|---|
| Swan Glaciers | 3 | Glaciers and perennial snowfields |

**Oregon**

**Cascade Range**

| | | |
|---|---|---|
| Bend Glacier | 3 | Glaciers and perennial snowfields |
| Clark Glacier | 2 | Perennial snowfields only |
| Collier Glacier | 2 | Glaciers only |
| Diller Glacier | 2 | Glaciers and perennial snowfields |
| Glisan Glacier | 2 | Glaciers and perennial snowfields |
| Ladd Glacier | 4 | Glaciers and perennial snowfields |
| Langille Glacier | 5 | Glaciers and perennial snowfields |
| Newton Clark Glacier | 3 | Glaciers and perennial snowfields |
| Palmer Glacier | 2 | Perennial snowfields only |
| Prouty Glacier | 3 | Glaciers and perennial snowfields |
| Renfrew Glacier | 2 | Glaciers and perennial snowfields |
| Russell Glacier | 2 | Glaciers only |
| Sandy Glacier | 4 | Glaciers and perennial snowfields |
| Skinner Glacier | 4 | Perennial snowfields only |
| Waldo Glacier | 3 | Glaciers only |
| White River Glacier | 2 | Glaciers and perennial snowfields |
| Whitewater Glacier | 3 | Glaciers only |
| Zigzag Glacier | 3 | Glaciers and perennial snowfields |

**Washington**

**Cascade Range-Northern**

| | | |
|---|---|---|
| Borealis Glacier | 4 | Glaciers only |
| Buckner Glacier | 2 | Glaciers only |
| Butterfly Glacier | 4 | Glaciers only |
| Colchuck Glacier | 2 | Glaciers only |
| Company Glacier | 3 | Glaciers only |
| Cool Glacier | 2 | Glaciers and perennial snowfields |
| Dana Glacier | 3 | Glaciers only |
| Dark Glacier | 3 | Glaciers only |
| Dome Glacier | 2 | Glaciers only |
| Douglas Glacier | 4 | Glaciers and perennial snowfields |
| Dusty Glacier | 2 | Glaciers and perennial snowfields |
| East Nooksack Glacier | 5 | Glaciers only |
| Entiat Glacier | 4 | Glaciers and perennial snowfields |
| Forbidden Glacier | 2 | Glaciers only |
| Fremont Glacier | 2 | Glaciers only |
| Goode Glacier | 2 | Glaciers only |

| | | |
|---|---|---|
| Hadley Glacier | 5 | Glaciers only |
| Hanging Glacier | 2 | Glaciers only |
| Hinman Glacier | 4 | Glaciers only |
| Honeycomb Glacier | 3 | Glaciers and perennial snowfields |
| Inspiration Glacier | 3 | Glaciers and perennial snowfields |
| Isella Glacier | 2 | Glaciers and perennial snowfields |
| Jerry Glacier | 2 | Glaciers only |
| Kimtah Glacier | 3 | Glaciers only |
| LeConte Glacier | 7 | Glaciers and perennial snowfields |
| Lyall Glacier | 2 | Perennial snowfields only |
| Mazama Glacier | 3 | Glaciers and perennial snowfields |
| McAllister Glacier | 2 | Glaciers only |
| Middle Cascade Glacier | 2 | Glaciers only |
| Neve Glacier | 3 | Glaciers only |
| No Name Glacier | 5 | Glaciers and perennial snowfields |
| Nohokomeen Glacier | 2 | Glaciers only |
| North Klawatti Glacier | 2 | Glaciers and perennial snowfields |
| Pilz Glacier | 3 | Glaciers and perennial snowfields |
| Price Glacier | 4 | Glaciers only |
| Ptarmigan Glacier | 2 | Glaciers and perennial snowfields |
| Queest-alb Glacier (not official) | 3 | Glaciers and perennial snowfields |
| Rainbow Glacier | 3 | Glaciers and perennial snowfields |
| Redoubt Glacier | 2 | Glaciers only |
| Richardson Glacier | 2 | Glaciers only |
| S Glacier | 3 | Glaciers only |
| Sandalee Glacier | 4 | Glaciers only |
| Scimitar Glacier | 3 | Glaciers only |
| Sholes Glacier | 4 | Glaciers only |
| Sitkum Glacier | 4 | Glaciers and perennial snowfields |
| Snow Creek Glacier | 2 | Perennial snowfields only |
| South Cascade Glacier | 2 | Glaciers only |
| Spider Glacier | 2 | Glaciers only |
| Suiattle Glacier | 2 | Glaciers only |
| Sulphide Glacier | 2 | Glaciers only |
| Thunder Glacier | 3 | Glaciers only |
| Thunder Glacier | 2 | Glaciers only |
| White Chuck Glacier | 5 | Glaciers and perennial snowfields |
| White Salmon Glacier | 2 | Glaciers only |
| Wyeth Glacier | 3 | Glaciers and perennial snowfields |

**Cascade Range-Southern**

| | | |
|---|---|---|
| Adams Glacier | 4 | Glaciers and perennial snowfields |
| Avalanche Glacier | 2 | Glaciers only |
| Conrad Glacier | 3 | Glaciers and perennial snowfields |
| Cowlitz Glacier | 2 | Glaciers and perennial snowfields |
| Crescent Glacier | 2 | Glaciers and perennial snowfields |
| Flett Glacier | 6 | Glaciers and perennial snowfields |
| Fryingpan Glacier | 5 | Glaciers and perennial snowfields |
| Gotchen Glacier | 2 | Glaciers and perennial snowfields |
| Kautz Glacier | 2 | Glaciers and perennial snowfields |

| | | |
|---|---|---|
| Klickitat Glacier | 2 | Glaciers only |
| Lava Glacier | 3 | Glaciers and perennial snowfields |
| McCall Glacier | 6 | Glaciers and perennial snowfields |
| Meade Glacier | 5 | Perennial snowfields only |
| North Mowich Glacier | 2 | Glaciers and perennial snowfields |
| Ohanapecosh Glacier | 6 | Glaciers and perennial snowfields |
| Paradise Glacier | 3 | Glaciers and perennial snowfields |
| Pinnacle Glacier | 3 | Glaciers and perennial snowfields |
| Puyallup Glacier | 2 | Glaciers and perennial snowfields |
| Pyramid Glacier | 4 | Glaciers and perennial snowfields |
| Russell Glacier | 2 | Glaciers only |
| Sarvant Glaciers | 4 | Glaciers and perennial snowfields |
| South Mowich Glacier | 2 | Glaciers only |
| South Tahoma Glacier | 2 | Glaciers and perennial snowfields |
| Success Glacier | 2 | Glaciers and perennial snowfields |
| Van Trump Glacier | 10 | Glaciers and perennial snowfields |
| White Salmon Glacier | 2 | Glaciers only |
| Whitman Glacier | 5 | Glaciers and perennial snowfields |
| Wilson Glacier | 3 | Glaciers and perennial snowfields |

**Olympic Mountains**

| | | |
|---|---|---|
| Blue Glacier | 2 | Glaciers only |
| Cameron Glaciers | 4 | Glaciers and perennial snowfields |
| Carrie Glacier | 2 | Glaciers only |
| Eel Glacier | 2 | Glaciers only |
| White Glacier | 2 | Glaciers only |

**Wyoming**
**Teton Range**

| | | |
|---|---|---|
| Middle Teton Glacier | 2 | Glaciers and perennial snowfields |
| Triple Glaciers | 3 | Glaciers only |

**Wind River Range**

| | | |
|---|---|---|
| Bull Lake Glacier | 3 | Glaciers and perennial snowfields |
| Dinwoody Glacier | 2 | Glaciers only |
| Dinwoody Glaciers | 3 | Glaciers and perennial snowfields |
| Grasshopper Glacier | 3 | Glaciers only |
| Harrower Glacier | 2 | Perennial snowfields only |
| Helen Glacier | 3 | Glaciers only |
| Lower Fremont Glacier | 4 | Glaciers and perennial snowfields |
| Mammoth Glacier | 2 | Glaciers and perennial snowfields |
| Minor Glacier | 2 | Glaciers and perennial snowfields |
| Sacagawea Glacier | 4 | Glaciers and perennial snowfields |
| Sourdough Glacier | 2 | Glaciers and perennial snowfields |
| Stroud Glacier | 3 | Glaciers and perennial snowfields |
| Twins Glacier | 2 | Glaciers and perennial snowfields |
| Upper Fremont Glacier | 2 | Glaciers and perennial snowfields |

**A3 Labelling errors in the U.S. Geographic Names Information System**

**Table A3**. List of officially named glaciers where we identified an issue with the glacier name
on the 1:24000 U.S. Geological Survey topographical maps (Fountain et al., 2017). Names come
from the Geographic Names Information System (https://www.usgs.gov/tools/geographic-
names-information-system-gnis). The 'Issue' column lists the type of issue identified. 'Not
labeled' indicates the feature was present but not labeled, 'Misidentified' indicates the wrong
feature was labeled, and 'Label unclear' means the location of the label is not clearly associated
with a specific glacier.

| State/Region/Glacier Name | Issue |
|---|---|
| **Colorado** | |
| **Front Range** | |
| Arikaree Glacier | Not labeled |
| Navajo Glacier | Not labeled |
| **Oregon** | |
| **Cascade Range** | |
| Carver Glacier | Misidentified |
| Milk Creek Glacier | Not labeled |
| **Washington** | |
| **Cascade Range-Northern** | |
| S Glacier | Label unclear |
| Snow Creek Glacier | Label unclear |
| South Glacier | Not labeled |
| **Cascade Range-Southern** | |
| No Name Glacier | Not labeled |
| Stevens Glacier | Not labeled |
| **Wyoming** | |
| **Wind River Range** | |
| Dinwoody Glaciers | Label unclear |
| Fremont Glaciers | Label unclear |

**A4 Notes on imagery and interpretation challenges by State.**
This appendix, organized by US State, then by mountain range, summarizes the specific
imagery used, challenges encountered in feature identification and outline digitization. The
Selkowitz and Forster (2016) inventory is referred to as the SFI and the National Land Cover
Database inventory (Dewitz, 2019) is referred to as the NLCD.
**A4.1 California**
Imagery and DEMs used are listed in Tables A4, A5, A6.
**Cascade Range**
**Mount Shasta**

The 2020 black and white Maxar imagery was most useful because of the minimal
seasonal snow cover. The 2018 NAIP imagery was helpful in situations where the 2020
imagery was obscured by shadow, distortion, or misaligned, and when color was needed
to improve interpretation. The 2010 lidar DEM (Robinson, 2014; Table A4) was used to
create a multidirectional hillshade to improve perspective and interpretation (Figure A1).
The rock debris on the termini of most glaciers and rock debris on some of the upper
parts of the glaciers were challenging to interpret. It was hard to determine whether ice
was present under the debris and whether that ice is part of the active glacier. Spatial
patterns of debris, debris contrasts, and melt streams flowing from the debris were used to
estimate the glacier boundaries.

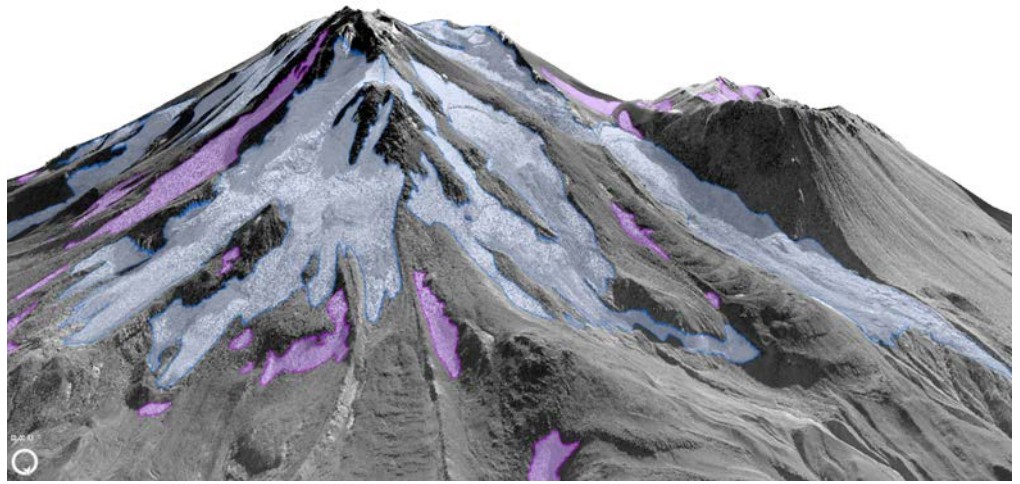

**Figure A1.** Mt. Shasta glaciers in bluish white, perennial snowfields/ice patches in lavender
draped over a 3D rendering created from 2010 lidar (Robinson, 2014).

**Sierra Nevada**

The 2014 NAIP imagery was the best imagery due to low snow cover. In some cases,
features were difficult to outline because of shadow or image quality. In these cases,
2013/2012 Google Earth imagery were used. Some glaciers were reclassified as rock
glaciers by Trcka (2020). These were re-examined and where we agreed they were
removed from the initial glacier inventory. Defining whether the feature was a glacier or
rock glacier was often difficult, see Colorado section for more discussion.

**Trinity Alps**

The 2018 imagery was the best for the least snow cover. Justin Garwood (Garwood et al.,
2020) provided outlines for two glaciers, Grizzly and Salmon. The area of the most recent
outline of the Salmon Glacier was < 0.01 km$^2$ and was not included in this inventory. By
2018 all of the other features mapped by the USGS (Fountain et al., 2017) were less than
0.01 km$^2$ or had disappeared. An additional feature was added based on the 2016 NLCD
(Jin et al., 2019).

**Table A4.** List of NAIP imagery used for outlining glaciers and perennial snowfields in
California. 'Date' is the start and end date for flights covering the glaciated portions of the NAIP
image. In some cases, flights were completed in a single day.

| Region/Year/Filename | County | Date (Year-M-D) |
|---|---|---|
| **Cascade Range** | | |
| 2014 | | |
| ortho_1-1_1n_s_ca089_2014_1.sid | Shasta | 2014-07-13 |
| ortho_1-1_1n_s_ca093_2014_1.sid | Siskiyou | 2014-06-23 to 2014-07-18 |
| 2018 | | |
| ortho_1-1_hn_s_ca093_2018_1.sid | Siskiyou | 2018-07-21 to 2018-09-25 |
| **Sierra Nevada** | | |
| 2014 | | |
| ortho_1-1_1n_s_ca019_2014_1.sid | Fresno | 2014-07-23 to 2014-08-23 |
| ortho_1-1_1n_s_ca027_2014_1.sid | Inyo | 2014-07-23 to 2014-08-23 |
| ortho_1-1_1n_s_ca039_2014_2.sid | Madera | 2014-07-18 to 2014-08-15 |
| ortho_1-1_1n_s_ca051_2014_1.sid | Mono | 2014-07-17 to 2014-08-15 |
| ortho_1-1_1n_s_ca107_2014_1.sid | Tulare | 2014-08-23 to 2014-08-23 |
| **Trinity Alps** | | |
| 2018 | | |
| ortho_1-1_hn_s_ca093_2018_1.sid | Siskiyou | 2018-07-21 to 2018-09-25 |

**Table A5**. List of dates of the Maxar imagery used for outlining glaciers and perennial
snowfields in California.

| Region/ Date (Year-M-D) |
|---|
| **Cascade Range** |
| 2020-10-05 |

**Table A6**. List of U.S. Geological Survey digital elevation models used for outlining glaciers
and perennial snowfields in California.

| Filename | Date | Citation | URL |
|---|---|---|---|
| ds852_lidar | 2010 | Robinson (2014) | https://pubs.er.usgs.gov/publication/ds852 |

**A4.2 Colorado**
The 2015 NAIP was generally free of seasonal snow. Where it persisted at the terminus of a few
glaciers, images for the same year in Google Earth aided perimeter interpretation. Imagery used

are listed in Table A7.

**Elk Mountains**

No features were mapped in the Elk Mountains by the USGS (Fountain et al., 2017). One

glacier and four perennial snowfields were added from the SFI.

**Front Range**

The most recent inventory for the Front Range was Hoffman et al. (2007), which used

aerial photographs to map the 2001 extent of glaciers. Many features in the Front Range

are difficult to classify. The issue is the difference between a glacier or perennial

snowfield and a rock glacier. Those that are part of the rock glacier are deleted from the

glacier inventory. Those that seem to be separate from rock glaciers are retained. This is a

judgement call. From a hydrological point of view, if a snow-ice patch that is part of a

rock glacier was counted separately from a rock glacier, it is double counting a water

feature.

**Table A7.** List of NAIP imagery used for outlining glaciers and perennial snowfields in

Colorado. 'Date' is the start and end date for flights covering the glaciated portions of the NAIP

image. In some cases, flights were completed in a single day.

| Region/Year/Filename | County | Date (Year-M-D) |
|---|---|---|
| **Elk Mountains** | | |
| 2015 | | |
|   ortho_1-1_1n_s_co051_2015_1.sid | Gunnison | 2015-09-10 to 2015-09-11 |
| **Front Range** | | |
| 2015 | | |
|   ortho_1-1_1n_s_co013_2015_1.sid | Boulder | 2015-08-25 to 2015-09-20 |
|   ortho_1-1_1n_s_co049_2015_1.sid | Grand | 2015-08-25 to 2015-09-20 |
|   ortho_1-1_1n_s_co057_2015_1.sid | Jackson | 2015-09-09 |
|   ortho_1-1_1n_s_co069_2015_1.sid | Larimer | 2015-08-25 to 2015-09-09 |
| **Gore Range** | | |
| 2015 | | |
|   ortho_1-1_1n_s_co037_2015_1.sid | Eagle | 2015-09-10 |
| **Medicine Bow Mountains** | | |
| 2015 | | |
|   ortho_1-1_1n_s_co057_2015_1.sid | Jackson | 2015-09-09 |
| **Park Range** | | |
| 2015 | | |
|   ortho_1-1_1n_s_co057_2015_1.sid | Jackson | 2015-09-09 |
| **San Miguel Mountains** | | |
| 2015 | | |
|   ortho_1-1_1n_s_co033_2015_1.sid | Dolores | 2015-09-11 |
|   ortho_1-1_1n_s_co091_2015_1.sid | Ouray | 2015-09-11 |
|   ortho_1-1_1n_s_co111_2015_1.sid | San Juan | 2015-09-12 |

    **Sawatch Range**
      2015

| Region/Year/Filename | County | Date |
|---|---|---|
| ortho_1-1_1n_s_co037_2015_1.sid | Eagle | 2015-09-10 |
| ortho_1-1_1n_s_co097_2015_1.sid | Pitkin | 2015-09-10 to 2015-09-11 |

**A4.3 Idaho**

The imagery quality was generally snow free. Of the glacier mapped by the USGS (Fountain et
al., 2017) only two remain and are classified as perennial snowfields. The Borah Glacier was
officially named in 2021 (U.S. Board of Geographic Names), but is < 0.01 km$^2$, and is not
included in the inventory. Table A8 lists the imagery used.

**Table A8.** List of NAIP imagery used for outlining glaciers and perennial snowfields in Idaho.
'Date' is the start and end date for flights covering the glaciated portions of the NAIP image. In
some cases, flights were completed in a single day.

| Region/Year/Filename | County | Date (Year-M-D) |
|---|---|---|
| **Sawtooth Range** | | |
|   2013 | | |
|     ortho_1-1_hn_s_id015_2013_1.sid | Boise | 2013-09-07 |
|   2015 | | |
|     ortho_1-1_1n_s_id013_2015_1.sid | Blaine | 2015-07-30 |
|     ortho_1-1_1n_s_id015_2015_1.sid | Boise | 2015-09-08 to 2015-09-09 |
|   2019 | | |
|     ortho_1-1_hn_s_id037_2019_1.sid | Custer | 2019-07-25 to 2019-08-26 |

**A4.4 Montana**

Inage quality varied between mountain ranges due to differences in snow cover. Tables A9 and
A10 list the imagery used.

**Beartooth-Absaroka Range**
The 2015 NAIP imagery was the best overall imagery due to the least snow, but Google
Earth was occasionally used as well. Google Earth had imagery dated to 9/11/2015; often
with less seasonal snow than the NAIP imagery. To counter any mismatch in projection,
outlines digitized in Google Earth were imported to ArcGIS and projected to match the
NAIP projection.
**Bitterroot Range**
No features were mapped in the Bitterroot Range by the USGS (Fountain et al., (2017).
One glacier and three perennial snowfields were added based on the NLCD.

**Cabinet Range**

The USGS mapped four features $\geq 0.01$ km$^2$ (Fountain et al., 2017).  Inspection of the 2015 only one was $\geq 0.01$ km$^2$. Seven glaciers and perennial snowfields were added; five were identified in our initial inventory, the other two were identified by the SFI and NLCD, respectively. All were less than 0.05 km$^2$.

**Crazy Mountains**
The 2013 NAIP imagery was the best imagery available and included limited seasonal snow. The 2019 Maxar imagery had too much seasonal snow.

**Lewis Range (Glacier National Park)**
The most recent published glacier inventory is a 2015 USGS inventory (Fagre et al., 2017). They outlined the main-body of named-glaciers using 2015 Maxar imagery. We digitized the outlines of all glaciers and perennial snowfields using 2015 Maxar imagery where available. Elsewhere, 2015 and 2013 NAIP imagery were used; both years had lots of seasonal snow cover. Two major glaciers, Blackfoot (Figure A2) and Harrison (Figure A3) glaciers, separated into pieces as it retreated since it was originally mapped by the USGS (Fountain et al., 2007).

**Madison Range**
The 2013 NAIP imagery was the only imagery used due to extensive snow in the other years. No glaciers or perennial snowfields were found. Of the two features $\geq 0.01$ km$^2$ mapped by the USGS (Fountain et al., 2017), the 2013 imagery showed that one feature is a rock glacier and the other was less than 0.01 km$^2$.

**Mission-Swan-Flathead Ranges**
Based on the least snow cover, the 2013 NAIP was better in the Mission and Flathead Ranges, and the 2015 NAIP was better in the Swan Range. No glaciers or perennial snowfields remain in the Flathead Range.

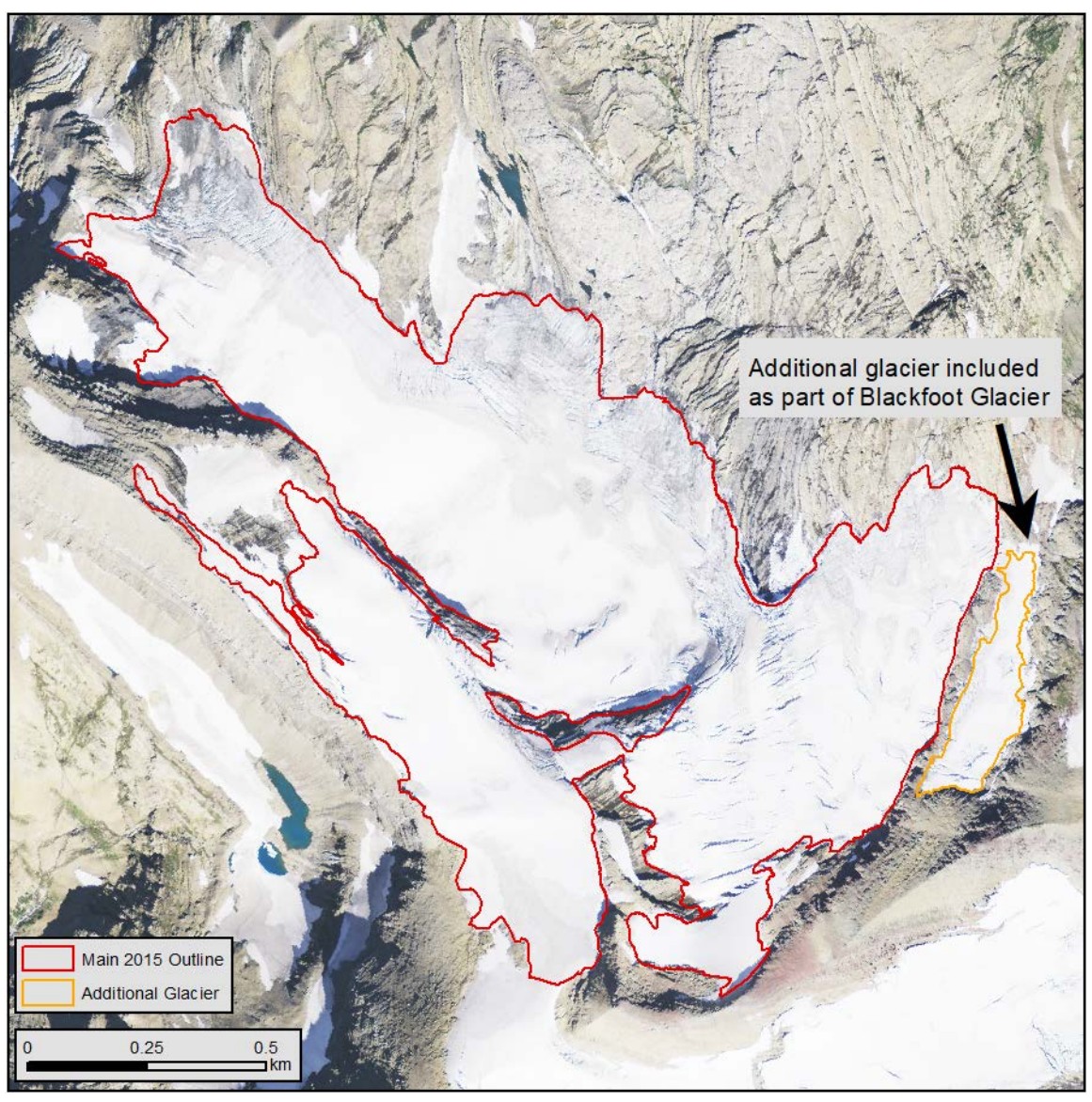

**Figure A2.** The updated (2015) outlines for the Blackfoot Glacier including the main glacier
body (red) and the additional smaller glacier (orange). Base image from the NAIP taken in 2013.

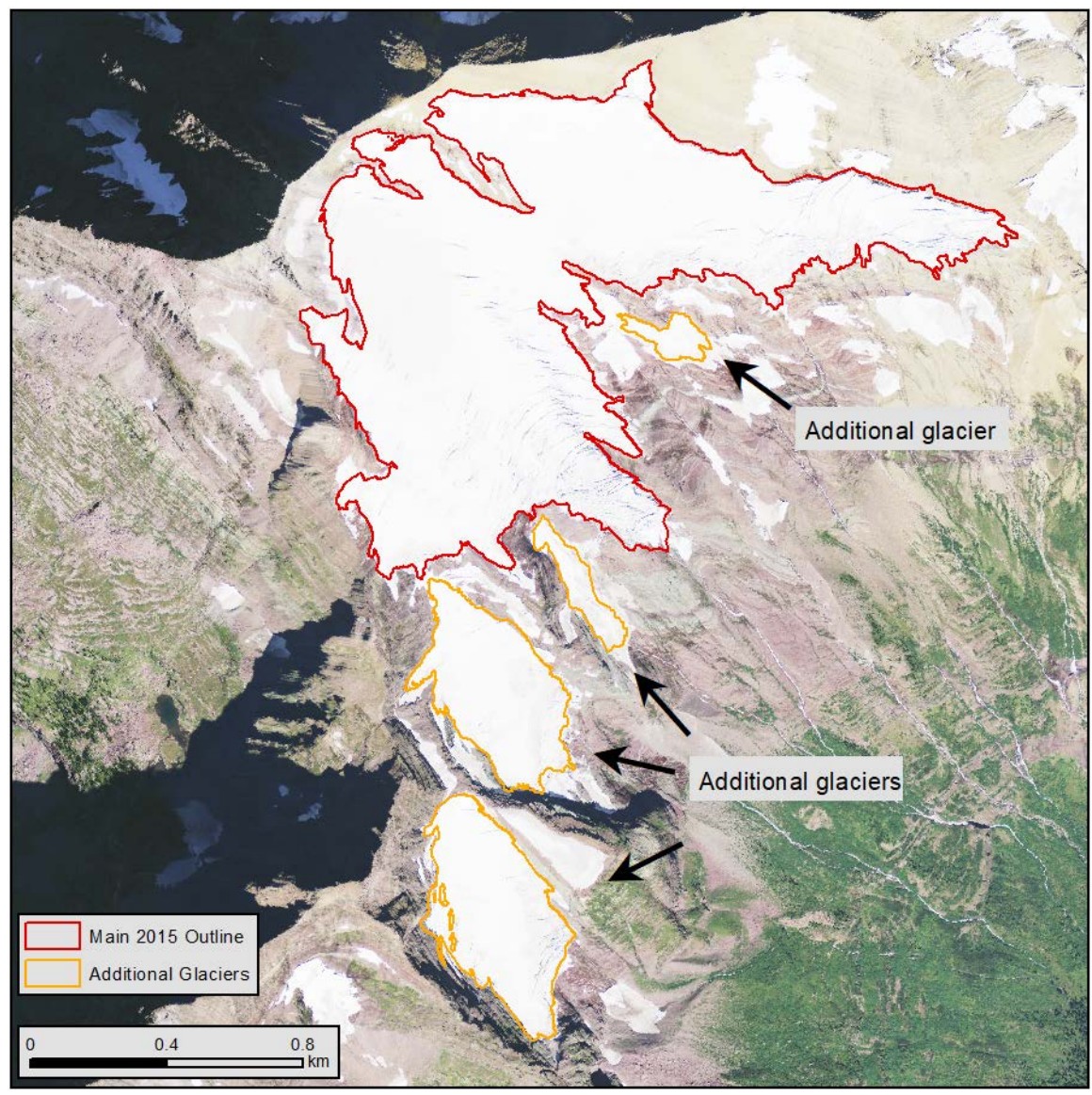

**Figure A3.** The updated (2015) outlines for Harrison Glacier including the main glacier body (red) and the additional smaller glaciers (orange). Base image from the NAIP taken in 2013.

**Table A9.** List of NAIP imagery used for outlining glaciers and perennial snowfields in Montana. 'Date' is the start and end date for flights covering the glaciated portions of the NAIP image. In some cases, flights were completed in a single day.

| Region/Year/Filename | County | Date (Year-M-D) |
|---|---|---|
| **Beartooth Mountains-Absaroka Range** | | |
| 2013 | | |
| ortho_1-1_1n_s_mt067_2013_1.sid | Park | 2013-08-05 to 2013-09-11 |

2015
ortho_1-1_1n_s_mt009_2015_1.sid   Carbon      2015-08-10 to 2015-09-07

| | | | |
|---|---|---|---|
| 2015 | | | |
| ortho_1-1_1n_s_mt009_2015_1.sid | Carbon | 2015-08-10 to 2015-09-07 |
| ortho_1-1_1n_s_mt067_2015_1.sid | Park | 2015-08-19 to 2015-09-11 |
| ortho_1-1_1n_s_mt095_2015_1.sid | Stillwater | 2015-08-10 to 2015-09-07 |

**Bitterroot Range**
  2013

| | | |
|---|---|---|
| ortho_1-1_1n_s_mt001_2013_1.sid | Beaverhead | 2013-08-04 |

  2015

| | | |
|---|---|---|
| ortho_1-1_1n_s_mt081_2015_2.sid | Ravalli | 2015-10-06 to 2015-11-07 |

**Cabinet Mountains**
  2015

| | | |
|---|---|---|
| ortho_1-1_1n_s_mt053_2015_2.sid | Lincoln | 2015-09-11 to 2016-08-15 |

**Crazy Mountains**
  2013

| | | |
|---|---|---|
| ortho_1-1_1n_s_mt067_2013_1.sid | Park | 2013-08-05 to 2013-09-11 |
| ortho_1-1_1n_s_mt097_2013_1.sid | Sweet Grass | 2013-08-31 to 2013-09-10 |

  2015

| | | |
|---|---|---|
| ortho_1-1_1n_s_mt067_2015_1.sid | Park | 2015-08-19 to 2015-09-11 |

**Lewis Range**
  2013

| | | |
|---|---|---|
| ortho_1-1_1n_s_mt029_2013_1.sid | Flathead | 2013-08-21 to 2013-09-01 |
| ortho_1-1_1n_s_mt035_2013_1.sid | Glacier | 2013-08-21 to 2013-09-01 |

  2015

| | | |
|---|---|---|
| ortho_1-1_1n_s_mt029_2015_2.sid | Flathead | 2015-09-30 to 2016-10-21 |
| ortho_1-1_1n_s_mt035_2015_2.sid | Glacier | 2015-10-14 to 2016-08-21 |

**Mission Range-Swan-Flathead Ranges**
  2013

| | | |
|---|---|---|
| ortho_1-1_1n_s_mt029_2013_1.sid | Flathead | 2013-08-21 to 2013-09-01 |
| ortho_1-1_1n_s_mt063_2013_1.sid | Missoula | 2013-09-01 |

  2015

| | | |
|---|---|---|
| ortho_1-1_1n_s_mt047_2015_2.sid | Lake | 2015-09-12 to 2016-08-15 |
| ortho_1-1_1n_s_mt063_2015_2.sid | Missoula | 2015-09-12 to 2016-08-16 |

**Table A10.** List of dates of the Maxar imagery used for outlining glaciers and perennial
snowfields in Montana.

**Region/ Date (Year-M-D)**
  **Lewis Range**
    2015-08-22
    2015-09-01
    2015-09-12
    2015-09-25
    2019-08-20

## A4.5 Oregon

Tables A11, A12, and A13 list the imagery and DEM used.

### Cascade Range

Seasonal snow cover was commonly present when this range was imaged by any of the sensors making it difficult to find suitable imagery.

**Mount Hood**
The most recent glacier outlines for Mt. Hood were based on 2015 and 2016 Maxar color imagery with interpretation aid using Google Earth. Due to seasonal snow some professional judgement was required in places.

**Mount Jefferson**
The 2018 NAIP had extensive seasonal snow and was generally only useful near the terminus of some glaciers. Used 2018 Maxar imagery that showed little seasonal snow, but a little cloudy that masked a bit of Whitewater Glacier. Also used Google Earth to help interpret some of the features.

**Three Sisters**
Maxar 2018 imagery was used, but the image was stretching along the feature's headwall and for that segment of the outline 2018 NAIP imagery was used. Two versions of the Maxar imagery for the same day are available, one color, one black and white. Color was georectified but suffered stretching along some headwalls. A light early season snowfall occurred before the Maxar image and the snow accumulated in some places just enough to obscure the surface. So, the glacier or snow patch outline was the minimum of the two images with occasional interpolation across the snowy surface to the nearest glacier edge.

**Mount Thielsen**
The Lathrop Glacier was named in 1981. At the time of the USGS mapping and now it is <0.01 km$^2$, and not counted as part of the inventory. Furthermore, Lathrop Glacier has been known to disappear in some years and therefore fails the definition of a glacier.

### Wallowa Mountains

No NAIP imagery was useful and Maxar did not image this region. We used the 8/30/2013 image from Google Earth, which was excellent with little snow. Features were digitized in Google Earth and then imported into ArcGIS. Because we used NAIP as the base imagery, we revised the outline from the projection in WGS84 (Google Earth) to NAD83 UTM Zone 11 (NAIP).

**Table A11.** List of NAIP imagery used for outlining glaciers and perennial snowfields in Oregon. 'Date' is the start and end date for flights covering the glaciated portions of the NAIP

image. In some cases, flights were completed in a single day.

| Region/Year/Filename | County | Date (Year-M-D) |
|---|---|---|
| **Cascade Range** | | |
| 2014 | | |
| ortho_1-1_1n_s_or017_2014_1.sid | Deschutes | 2014-09-01 |
| ortho_1-1_1n_s_or027_2014_1.sid | Hood River | 2014-08-27 to 2014-09-05 |
| ortho_1-1_1n_s_or039_2014_1.sid | Lane | 2014-09-01 |
| 2016 | | |
| ortho_1-1_1n_s_or027_2016_1.sid | Hood River | 2016-08-04 |
| 2017/2018 | | |
| ortho1-1_hn_s_or017_2017_2018_1.sid | Deschutes | 2018-07-28 |
| **Wallowa Mountains** | | |
| 2014 | | |
| ortho_1-1_1n_s_or063_2014_1.sid | Wallowa | 2014-10-05 |

**Table A12**. List of dates of the Maxar imagery used for outlining glaciers and perennial
snowfields in Oregon.

| Region/ Date (Year-M-D) |
|---|
| **Cascade Range** |
| 2015-08-20 |
| 2015-09-11 |
| 2015-10-05 |
| 2016-09-10 |
| 2018-09-17 |
| 2020-09-20 |

**Table A13.** List of Oregon Department of Geology and Mineral Industries digital elevation
models used for outlining glaciers and perennial snowfields in Oregon.

| Filename | Date | URL |
|---|---|---|
| 2011_OLC_Deschutes | 2011 | gis.dogami.oregon.gov/maps/lidarviewer/ |

**A4.6 Washington**
The 2015 NAIP imagery was typically excellent with little snow cover, whereas the 2017 NAIP
had more snow and the 2019 imagery had lots of snow. For most outlines, 2015 NAIP imagery
was used. In some places, the 2017 NAIP imagery had less snow and was used instead. Maxar
imagery was of limited use and often wasn't better than the 2015 or 2017 NAIP. Tables A14,
A15, A16, list the imagery and DEMs used.

**Cascade –Northern**

The glaciers and perennial snowfields were previously inventoried by (Dick, 2013).

**Mount Baker**

The 2015 NAIP imagery was the best and had little seasonal snow. Google Earth 2009 and 2019 imagery were used to help interpretation. A multidirectional hillshade and 3-meter contour lines derived from a lidar DEM (Bard, 2017a); were used to help define flow divides between glaciers, debris covered-ice, and buried ice. There are notable differences between the NAIP imagery and DEM data, particularly in steep terrain, areas of dark shadow, and debris-covered areas. The DEM helped correct these positional errors and the benefit of supplying more information on surface texture.

Several buried-ice features were identified. The ice appeared to have decoupled from the active glacier. In a few cases, debris-covered ice is included in the glacier outline because the ice appears to be directly connected to the glacier, and there was evidence of movement.

**Dragontail Peak**

The USGS Geographic Names Information Service (GNIS) locates Snow Creek Glacier at a point on the edge of the southeast glacier (Fountain et al., 2007). In the 2015 imagery, the point is on bedrock, making it unclear which glacier the GNIS is naming. The USGS identifies both glaciers as Snow Creek Glacier. We labeled both glaciers as the Snow Creek Glacier.

**Glacier Peak**

For the Glacier Peak region, a multidirectional hillshade and 3-m contour lines derived from a 2015 lidar DEM (Bard, 2017b) were used as a guide to define flow divides.

**Hurry-up Peak**

The point location of the South Glacier provided by the GNIS is over bedrock. We assume the point refers to the glacier located ~150 m to the north of the point.

**Cascade –Southern**

**Goat Rocks**

Imagery from 2015 was best, but had more snow than desired. Too much snow was present in 2017 but some ice was exposed. The 2019 imagery was too snowy for glacier digitization.

The outlines are almost entirely based on 2015 imagery, and a few on 2017, where needed. Used 2009 NAIP imagery to help define the headwalls at the Conrad, McCall, and Packwood glaciers. Heard  (2000) previously mapped the glacier perimeters. The maximum extent of the seasonal snow covering the terminal regions was not digitized. Typically digitized at scales of 1:600 to 1:800. Note that narrow arms of the snowfields were not typically digitized knowing that they would probably disappear a few days to a

653     week from the time of imagery.
**Mount Adams**
No suitable NAIP imagery was found, instead 2019 Maxar imagery was used. In addition
to the Maxar imagery, a multidirectional hillshade and 3-m contour lines derived from a
2016 lidar DEM (Bard 2019) were used as a guide when delignating flow divides.
Occasionally, 2009 Google Earth imagery was also useful. Extensive snow covered the
mountain when the 2016 lidar was flown masking some of the glacier termini. However,
the DEM was helpful in correcting the imagery where poorly aligned with the terrain.
Multiple buried-ice features were identified near the terminus of several glaciers where
ice appeared to have decoupled from the main active glacier. Large areas below the
glaciers (Mazama, Adams, and Pinnacle) likely have debris-covered ice. We focused on
the features which were likely to contain ice based on meltwater streams exiting near the
features and hummocky terrain which appeared to indicate melt. Ground-based images
from taken between 2014 to 2018 helped decision-making. The images were particularly
helpful in identifying a debris-covered ice cliff at Adams Glacier.
**Mount Rainier**
In general, the 2019 NAIP and the Maxar (2018-09-25) were used for the outlines.
Although the GNIS includes the Nisqually Icefall as a separate feature, we included the
icefall as part of the Nisqually Glacier (Figure A4).
**Mount St. Helens**
We used a GIS layer of geological mapping units that included snow and ice from the
USGS (David Sherrod, USGS written communication, 2021) to help guide our search.
The Crater Glacier (INV_ID E562842N5115499) was heavily debris covered, and
obscured by shadow in some areas.
**Olympic Mountains**
A 2015 inventory of the region was compiled because more recent imagery (NAIP and
Maxar) were not useful due to seasonal snow. Our updated inventory differs from that
published in Fountain et al. (2017) in two ways. First, they outlined and grouped the
glaciers and perennial snowfields according to watershed rather than individual glacier.
Their goal was to estimate glacier change relative to a previous study by Spicer (1986)
and had to follow Spicer's approach. Second, all outlines were rechecked and compared
to SFI and the NLCD resulting in minor changes.

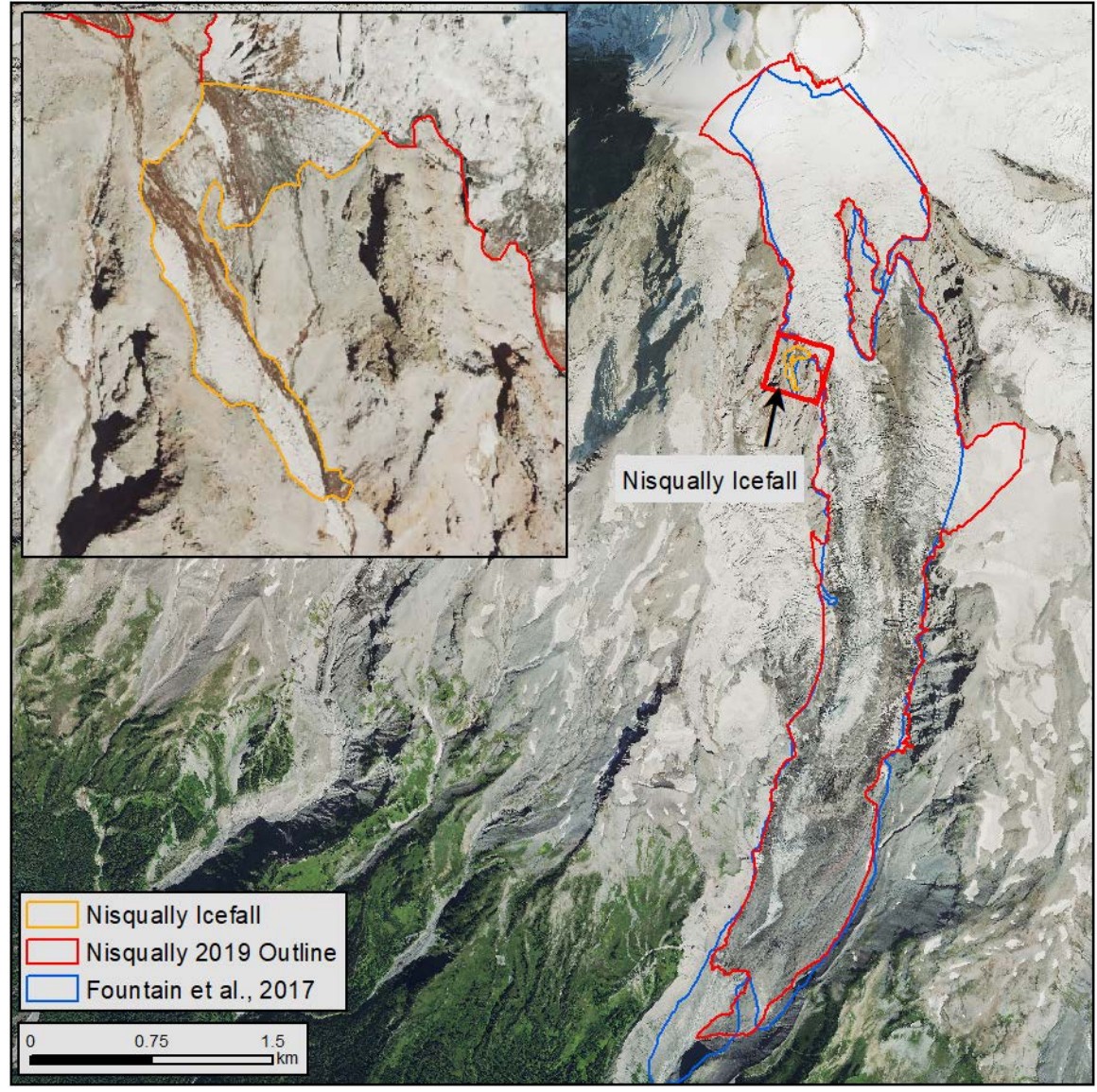

**Figure A4.** Image of the Nisqually Glacier and Icefall. The orange and red outlines are from the
updated inventory and the blue outline is from the USGS mapping (Fountain et al., 2007)
database. The base image is from the NAIP taken in 2019.

**Table A14.** List of NAIP imagery used for outlining glaciers and perennial snowfields in
Washington. 'Date' is the start and end date for flights covering the glaciated portions of the
NAIP image. In some cases, flights were completed in a single day. For 2006 the inspection date
was used, since the start and end dates were not provided.

| Region/Year/Filename | County | Date (Year-M-D) |
|---|---|---|
| **Cascade –Northern** | | |

2006
ortho_1-1_1n_s_wa007_2006_3.sid         Chelan          2006-07-01
2015
  ortho_1-1_1n_s_wa007_2015_1.sid       Chelan          2015-07-06 to 2015-09-23
  ortho_1-1_1n_s_wa033_2015_1.sid       King            2015-07-06 to 2015-09-27
  ortho_1-1_1n_s_wa037_2015_1.sid       Kittitas        2015-07-06 to 2015-09-23
  ortho_1-1_1n_s_wa047_2015_1.sid       Okanogan        2015-09-09 to 2015-09-11
  ortho_1-1_1n_s_wa057_2015_1.sid       Skagit          2015-07-06 to 2015-09-29
  ortho_1-1_1n_s_wa061_2015_1.sid       Snohomish       2015-07-06 to 2015-09-29
  ortho_1-1_1n_s_wa073_2015_1.sid       Whatcom         2015-09-10 to 2015-09-26
2017
  ortho_1-1_1n_s_wa007_2017_1.sid       Chelan          2017-10-03 to 2017-10-24
  ortho_1-1_1n_s_wa057_2017_1.sid       Skagit          2017-09-27 to 2017-10-05
  ortho_1-1_1n_s_wa073_2017_1.sid       Whatcom         2017-09-27 to 2017-10-05

**Cascade –Southern**
2015
  ortho_1-1_1n_s_wa041_2015_1.sid       Lewis           2015-07-15 to 2015-07-29
  ortho_1-1_1n_s_wa053_2015_1.sid       Pierce          2015-07-29
  ortho_1-1_1n_s_wa059_2015_1.sid       Skamania        2015-07-15 to 2015-09-12
  ortho_1-1_1n_s_wa077_2015_1.sid       Yakima          2015-07-15 to 2015-07-29
2019
  ortho_1-1_hn_s_wa053_2019_1.sid       Pierce          2019-08-26
  ortho_1-1_hn_s_wa059_2019_1.sid       Skamania        2019-08-06 to 2019-08-26

**Olympic Mountains**
2015
  ortho_1-1_1n_s_wa009_2015_1.sid       Clallam         2015-07-28 to 2015-09-12
  ortho_1-1_1n_s_wa031_2015_1.sid       Jefferson       2015-07-28 to 2015-09-12
  ortho_1-1_1n_s_wa045_2015_1.sid       Mason           2015-07-28 to 2015-08-19

**Table A15.** List of dates of the Maxar imagery used for outlining glaciers and perennial
snowfields in Washington.

**Region/ Date (Year-M-D)**
  **Cascade Range-Northern**
    2018-09-25
  **Cascade Range-Southern**
    2018-09-25
    2019-08-31
  **Olympic Mountains**
    2015-08-17
    2019-09-30

**Table A16.** List of U.S. Geological Survey digital elevation models used for outlining glaciers

and perennial snowfields in Washington. To access the data both the URL and specific identifier
are required.

| Region | Date | Citation | URL www.sciencebase.gov/catalog/item/ |
|---|---|---|---|
| Mt. Adams | 2016 | Bard (2019) | 5bc623b9e4b0fc368ebbe99a |
| Mt. Baker | 2015 | Bard (2017a) | 58518b0ee4b0f99207c4f12c |
| Glacier Peak | 2014-15 | Bard (2017b) | 57bf299ee4b0f2f0ceb7534e |

**A7 Wyoming**
**Wind River Range**
Tables A17 and A18 list the imagery used. The 2015 NAIP imagery had little snow in
contrast to 2019 imagery. Shadows are common in the 2015 imagery and can be very
dark. Occasionally the 2019 imagery was used to define the glacier-bedrock headwall
boundary. The 2019 Maxar imagery was essentially identical to the NAIP and because its
black and white not as useful. Imagery from 2017 and 2018 were a bit too snowy around
the glacier margin to be useful. The 2018-09-06 Maxar imagery covered the entire range,
with some clouds.

In the southern Wind River Range, a new snow dusting was often present, occasionally
making it difficult to outline snowfields and a few glaciers. Distinguishing seasonal snow
from perennial snow was a judgement call. If the snow was slightly discolored similar to
underlying rock/soil looking like the color was coming from underneath it was identified
as seasonal snow. Also, if many snow-free patches (a few square meters) pockmarked the
snow or if many rocks protruded through the snow, it was considered seasonal. A
perennial patch of snow appeared smooth and white, hiding underlying surface. Thin
snow cover on glacier ice appeared greyish in color and appeared smoother than the
surrounding ice-free landscape.

At Lower Fremont Glacier, a number of sizable ice patches appear down valley as if a
deposit of buried ice is present. However, there is no obvious connection to the glacier
itself.

The GNIS identified a single glacier as the Sacagawea Glacier, and two separate Fremont
Glaciers (Figure A5). By 2017 the single glacier had split into four glaciers. We chose to
label the largest glacier and the glacier to the southeast the Sacagawea Glacier. The other
two glaciers were labeled the Fremont Glaciers.

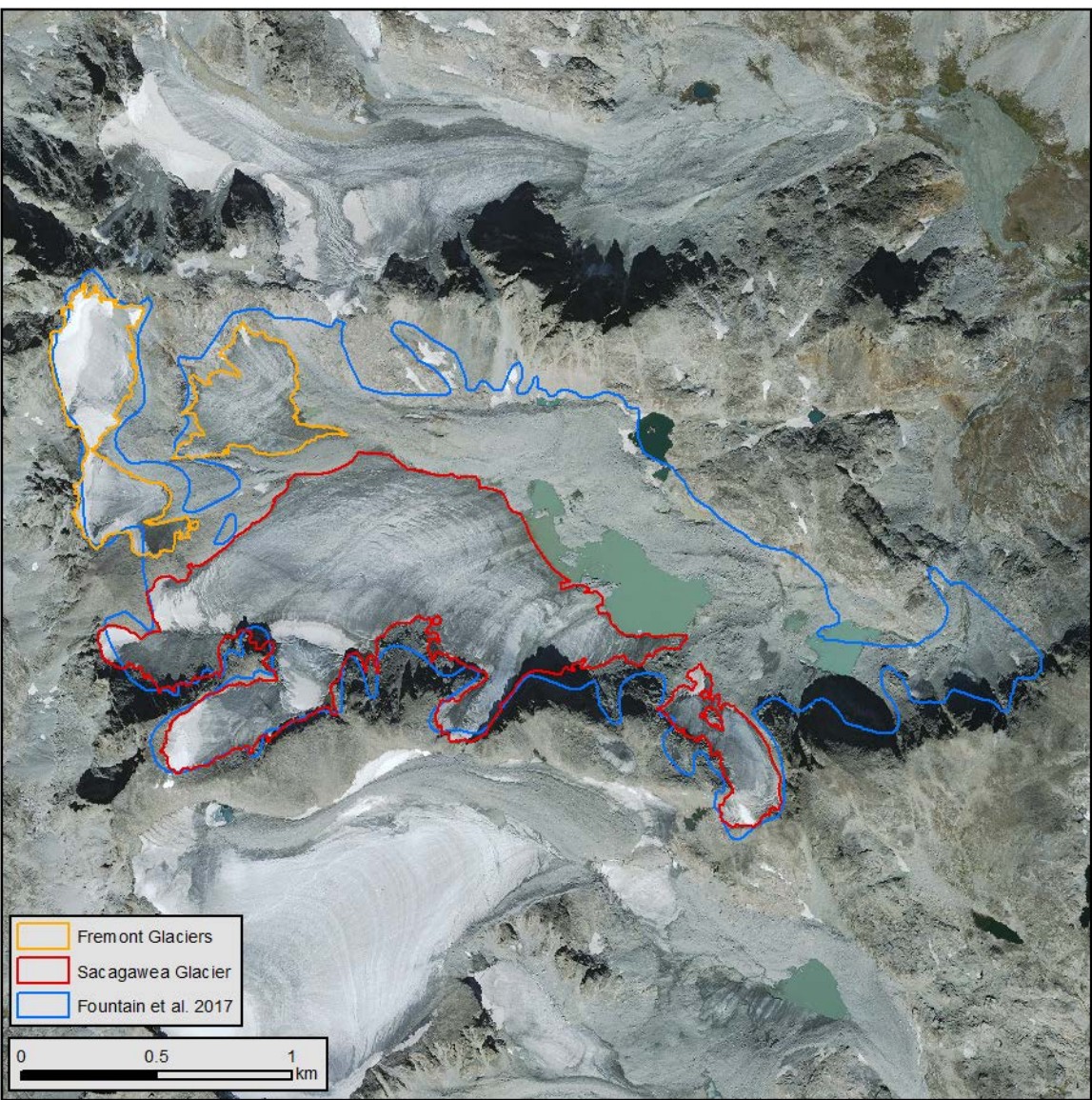

**Figure A5.** Image of Fremont Glaciers and Sacagawea Glacier showing the Sacagawea outline
from the Fountain et al. (2017) database (blue), our updated Fremont Glaciers outlines (orange),
and updated Sacagawea outlines (red). The base image is from the NAIP, taken in 2015.

**Table A17.** List of NAIP imagery used for outlining glaciers and perennial snowfields in
Wyoming. 'Date' is the start and end date for flights covering the glaciated portions of the NAIP
image. In some cases, flights were completed in a single day. For 2006 the inspection date was
used, since the start and end dates were not provided.

| Region/Year/Filename | County | Date (Year-M-D) |
|---|---|---|
| **Absaroka Range** | | |

| | | |
|---|---|---|
| 2006 | | |
| ortho_1-2_1n_s_wy029_2006_1.sid | Park | 2006-09-02 |
| 2015 | | |
| ortho_1-1_hn_s_wy013_2015_2.sid | Fremont | 2015-09-09 to 2015-10-13 |
| ortho_1-1_hn_s_wy029_2015_2.sid | Park | 2015-09-22 to 2015-10-13 |
| **Bighorn Mountains, WY** | | |
| 2015 | | |
| ortho_1-1_hn_s_wy019_2015_2.sid | Johnson | 2015-09-12 |
| **Teton Range** | | |
| 2006 | | |
| ortho_1-1_1n_s_wy039_2006_1.sid | Teton | 2006-09-02 |
| 2015 | | |
| ortho_1-1_hn_s_wy035_2015_2.sid | Sublette | 2015-09-09 to 2015-10-13 |
| ortho_1-1_hn_s_wy039_2015_2.sid | Teton | 2015-09-12 to 2015-09-22 |
| 2019 | | |
| ortho_1-1_hn_s_wy039_2019_1.sid | Teton | 2019-07-20 to 2015-09-22 |
| **Wind River Range** | | |
| 2006 | | |
| ortho_1-1_1n_s_wy035_2006_1.sid | Sublette | 2006-09-02 |
| 2015 | | |
| ortho_1-1_hn_s_wy013_2015_2.sid | Fremont | 2015-09-09 to 2015-10-13 |
| ortho_1-1_hn_s_wy035_2015_2.sid | Sublette | 2015-09-09 to 2015-10-13 |
| 2019 | | |
| ortho_1-1_hn_s_wy013_2019_1.sid | Fremont | 2019-07-20 to 2019-08-27 |
| ortho_1-1_hn_s_wy035_2019_1.sid | Sublette | 2019-08-15 to 2019-09-13 |

**Table A18.** List of dates of the Maxar imagery used for outlining glaciers and perennial
snowfields in Wyoming.

**Region/ Date (Year-M-D)**

Wind River Range

2018-09-06

**8. Author Contributions.**

Andrew G. Fountain was the principal investigator of the project, he wrote the proposal and
digitized glacier and snowfield outlines, analyzed the data, and led the writing of this report.
Bryce Glenn was the GIS expert responsible for the geographic format (e.g. projection,
attributes, database structure) and quality control. He digitized glacier and snowfield outlines,
analyzed the data, and helped write the report. Chris McNeil provided some of the imagery.

**9. Competing Interests.**

The authors declare that they have no conflict of interest.

**10. Acknowledgments.**

Hassan Basagic and Kristina Dick digitized the glaciers of Mt. Shasta and the North Cascade National Park, respectively. We greatly appreciate their help and expertise. Nathan Walker of the U.S. Forest Service provided a very helpful initial review of this report. We kindly acknowledge the funding from the US. Forest Service. Any use of trade, firm or product names is for descriptive purposes only and does not imply endorsement by the U.S. Government.

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
