# Peer review of "Inventory of glaciers and perennial snowfields of the conterminous USA"

_Earth System Science Data, 2022_

## Referee Comment (RC3)

**Comments to ESSD-2022-369**

**General comments**

The study by Fountain et al. is presenting the results of a new glacier inventory for the contiguous United States /without Alaska) as mapped from manual digitizing of orthorectified digital aerial imagery. Independent of my comments below, I want to congratulate the authors to this long overdue update and acknowledge the great effort that is visible here. I have a number of more general comments to a) the terminology, b) the 'layout of the 'paper' and c) the datasets as well as some more specific ones.

Starting with a), I am not very happy with mixing glaciers and perennial snowfields. First, snow fields should not be included in a glacier inventory and second, I think the definition applied here for 'perennial snowfields' is ambiguous. Of course, glaciers can be described as perennial snow (L82), as they originate from snow that survives the melting season over several years, but I think this is not the same as 'perennial snow fields' that are just composed of snow and firn and should thus not be included in a glacier inventory. Things get a bit complicated when ice patches (not moving glacier ice) - that might be completely covered by seasonal snow - are to be included but seasonal snow has to be excluded.

Given that seasonal and perennial snowfields are abundant in this region, that their separation is nearly impossible in many cases, and that the transition of a glacier to an ice patch is gradual, I suggest using a definition that is better suited for this environment. Leigh et al. (2019) have tried to sort this out with a scoring system that can be applied when very high-resolution images are available. I suggest testing it here and re-evaluate the assignment. Currently, a large number of the here assigned 'perennial snowfields' are actually glaciers, e.g. they show bare ice, deformed debris bands, lateral moraines and could be found 'above' units classified as glaciers (and in a few cases its also the other way round, some avalanche deposits in valley floors are classified as glaciers). One example is shown in Figure A10. I think there is no need to assign the class 'Perennial snowfield' to (the two parts of) Freemont glacier. This is still a usual glacier that is actually connected in its lower part (under a thick medial moraine) to the neighbouring Sacagawea Glacier.

I fully acknowledge the difficulties in performing such an assignment and that in many cases a clear assignment might even be impossible, but currently the number of real glaciers that would be removed from the sample when users exclude the perennial snowfield class is rather high and thus worth revisiting. Moreover, in some regions it seems that perennial snowfields (and even glaciers according to Leigh et al. 2019) have not been mapped. Once this is done, please add example images for the various cases in a multi-panel figure so that readers have a chance to follow the decisions. Maybe also a short note on the class 'Buried Ice': I would not use it. Include it with the glacier outline when it looks like glacier ice under debris-cover and leave it when not. None of the dataset users will do the reassignment, but all are aware and will understand that other interpretations might exist. So please decide as an analyst where to place the glacier outline and leave it with this.

Regarding point b), I also have the impression that the current draft reads more like an internal progress report rather than a paper. There is no problem with being short and to the point, but for example a discussion is completely missing, the information included in the attribute table is not presented, glacier characteristics to be included in a glacier inventory (e.g. minimum, mean, maximum and median elevation, or mean slope and aspect) are neither calculated nor presented and visualized and lots of information is listed in the Appendix without providing a good access (e.g. showing the image footprints) or mapping examples from the various regions with outline overlays to see the decisions taken (also in difficult cases). The text provided in the Appendix comes thus across as rather theoretical descriptors of image conditions and would in this form have a better place in the Supplemental Material. Some of the images in the Appendix, however, should be transferred to the main part and used to illustrate the methods. Please show outline overlays and annotate the images to guide the readers through the decisions made.

The digitizing of the new dataset has in general an excellent quality and is a clear improvement over the currently available datasets. However, as mentioned above, the assignment of 'this is a glacier' and 'this a perennial snowfield' seems a bit arbitrary and inconsistent at times. Moreover, some glaciers (and/or perennial snow fields) are seemingly missing. I have compiled a few examples at the end of this review for illustration and suggest revisiting the assignment of all perennial snowfields to really have all glaciers included in the glacier class. Please add an item Class_nr with 1 for glaciers and 2 for PS. I would also encourage the authors to calculate topographic information for each glacier entity, provide the data in the attribute table of the data file and add some selected illustrations of the dataset characteristics to the text (e.g. maps, scatterplots, bar charts, tables). A shape file providing image footprints (to see which outlines have been derived from which image) would be a most welcome asset.

**Specific comments**
L1: conterminous or (first n missing) or contiguous (as in L9)?
L26/27: What are the criteria to cite these publications? Not all of them are about stream flow.
L27/29 (and elsewhere): When referring to contemporary glaciers, I would use glacier instead of glacial (see Cogley et al. 2011)
L62/64: I suggest not naming it a report when it should be a paper, maybe use this study.
L67: Please give this part an individual subsection 2.1 (and Uncertainties in L132 to 2.2)
L82: How was the 0.01 km$^2$ size threshold applied before the digitizing?
L81-87: I suggest applying the classification system suggested by Leigh et al. (2019) to get a better handle on what is a perennial snowfield and what can be named a glacier.
L90: Shaded reliefs are often ambiguous. I suggest using a flow-direction grid to separate glacier complexes into individual entities.
L103: In fact, this IS a huge common problem.
L112 (and elsewhere): Please number all sections in the Supplemental Material and refer here also to this number.
L117: 'once part of the glacier': Couldn't this be checked against the previous inventory?
L132: Please give Uncertainties and individual subsection (2.2)
L142: digitizing
L148-170: As mentioned in the general comments, can you please illustrate with a Figure how these datasets (SFI and NLCF) look like and how the merging was done?
L172: Please add an analysis of glacier characteristics as derived from a DEM
L190: I suggest moving this table to the Supplemental Material and showing in the text only a figure (bar or pie chart). And remove the 'Buried ice' class. Either include or exclude it.
L205: I think also Table 2 has a better home in the supplemental Material. This is background information, there is little that can be learned from it.
L210: Please add a Discussion section
L262: Can you explain here why some of these are perennial snowfields while others are glaciers. It is not entirely clear, in particular not for the large ones.
Figures A3, A6 and A9 miss outline overlays. Where is Figure A8?
Leigh et al. (2019): https://doi.org/10.1017/jog.2019.50

**Image examples (World Imagery layer of the ESRI Basemap)**

Red: glaciers, yellow/orange: perennial snowfields, green: burried ice, blue: RGI6

x: this is also a glacier. Circle: these should be included, at least as perennial snowfields. Background image: ESRI.

[Figure]

x: glaciers rather than perennial snowfields, Circles: Missing. Background image: ESRI.

[Figure]

I would not include the green part. Background image: ESRI.

[Figure]

This is an avalanche deposit rather than a glacier. Maybe not even a perennial snowfield? Background image: ESRI.

[Figure]

Where is the debris covered part (yellow circle), why is this (x) not a perennial snowfield. Background image: ESRI.

[Figure]

The orange outlines should be glaciers rather than perennial snowfields. x looks like a rock glacier. Background image: ESRI.

[Figure]

The orange outlines (larger polygons) should be glaciers rather than perennial snowfields. Background image: Copernicus Sentinel-2 2020

[Figure]

The orange outlines should be glaciers rather than perennial snowfields. Background image: ESRI.

---

## Author Comment (AC1)

Author Responses in **bold italics**

The authors present an updated inventory of glacier, rock glaciers and perennial snowfields. This is a much appreciated and needed update since the currently available inventory is based on topographic maps from the 1980s or earlier. This inventory is available through the GLIMS data base also part of the current version of the Randolph Glacier Inventory (vers. 6.0).

The authors identify and digitise the glaciers based on very-high resolution aerial imagery (spatial resolution 1m or better) and use imagery available through Google Earth as additional information. The inventory is nicely illustrated and details of the different mountain ranges presented.

Overall, the work seems to be performed with care and outlines seem to be of high quality. However, the paper is written like a report and not a scientific paper, the description of the methodology is a bit thin and the accuracy assessment could be improved. Moreover, some additional analyses, like the comparison to the old inventory and the inclusion of typical inventory parameters would be very beneficial. Overall the manuscript needs some improvements and revisions but should ultimately be published.

> ***We appreciate the reviewer's efforts editing our manuscript. Wehave extensively revised the text to improve clarity and rigor.***

More details are presented below.

L16/17: All numbers of glacier areas should be presented along with their uncertainty ranges.

> ***Revised***

L17: What are buried-ice features? Do these include rock glaciers? As I understand you have a separate class "rock glaciers". However as these are not glaciers per se, these should also be treated separately (see more details below)

> ***The buried ice features are related to glaciers, probably dead ice adjacent to a glacier. They are not rock glaciers. See response your more detailed comment below.***

L32: A nice paper which showed this (although for Asia not for the US) is Pritchard (2019). A consideration of Huss and Hock (2018) would also nicely fit here.

> ***Good suggestions, thanks!***

L39ff: This is a bit confusing: You may want to mention already here that an inventory is existing (and refer to the reference), but that the inventory is outdated.

> ***Revised***

L 62: This should be rather a scientific paper and not a report.

*Revised*

L 69ff: The info how the utilised images were orthorectified is missing but should be included along with some info about the accuracy of the orthoimages. How were the google earth images used? Were they co-registered to the aerial imagery? How well does the geolocation fit to the old maps resp. the former inventory?

*Revised*

L 82: Provide a rational for the **selected** threshold.

*We're unclear the reason for the question. The resolution is that of the imagery and 1m or less is more than sufficient for digitizing outlines.*

L86: It would be beneficial to attempt to distinguish between perennial snow and ice. I think it should be possible, e.g. if at one scene only bare ice is visible than it is an ice patch. But I do not know the images and it might be difficult or introduce to much subjectivity.

*This would be lots of work to go through all of the images again to determine which are ice patches for what we feel would be little gain. From our experience there were only a few ice patches. And as the reviewer suggest distinguishing the two will be difficult largely because of the dark shadow that often blankets ice patches.*

Paragraph L94ff: I ask the authors to be more specific with the challenges and present some example figures in the main text and not only in the regions. It is e.g. well known that debris-covered ice is partially difficult to identify and the delineation subject to the operator (and even the operation is not consistent) as nicely illustrated by Paul et al. (2013). How did you deal with the problem?

*Figures moved from the Appendix to the main text. Regarding debris-covered glaciers difficult to outline were largely limited to the volcanoes of the Cascade Range. We had local knowledge in many places. In others, the independent digitizations by the authors and others helped to define uncertainty (explained in Uncertainty). The text was revised to address this issue.*

Paragraph L103ff: This section needs more depth and more detailed information. As correctly mentioned, it is not trivial to separate glaciers from rock glaciers. I recommend moving some of the text from the different regions to the main text (e.g. L271f, l309ff) and be more specific. Moreover, rock glaciers are commonly associated with permafrost. I know that opinions diverge, but this should at least be mentioned, and you need to consider the guidelines by the IPA Action Group Rock glacier inventories and kinematics (RGIK, 2022). You may also think about overlaying existing permafrost modelling results with the rock glacier inventory. Snow fields above rock glaciers are usually not considered as a part of rock glaciers. In general, the literature should be better considered here (e.g. Ostrem, 1970,

Janke 2007, Mölg et al. 2018, Charbonneau and Smith, 2018, Janke and Bolch 2021 but there are several other relevant papers).

> ***Some of the text from the appendix was moved to the main article per the reviewer's suggestion to better explain how we distinguished glaciers from rock glaciers. We read the suggested citations and adjusted the text. We found the RGIK (2022) to be most useful.***

L113ff: I appreciate that you include buried ice adjacent to glaciers. However, the terminology is a bit unclear. Are these ice-cored moraines (see e.g. Lukas 2011)? Or ice identified in the forefield of glaciers? Interesting papers in this regard might be Lukas et al. (2005), Bolch et al. (2019).

> ***Thank you for these references, we found Bolch et al (2019) to be helpful.***

L127f: More details are needed. What were the difficulties? How where they overcome (e.g. in case of debris cover)? How many of the outlines had to be adjusted?

> ***Text was adjusted. The issues regarding debris cover was addressed in the Uncertainty section. We didn't record how many outlines had to be adjusted. As mentioned in the uncertainty section, the adjustment was used in the uncertainty estimate.***

L132ff (Uncertainty). A more rigorous uncertainty assessment is needed. The digitisation error itself might be small if the glaciers are well identifiable, but many are not. The image resolution had also an influence (cf. Paul et al. 2013 as cited earlier). You may think to use a buffer (e.g. Granshaw and Fountain, 2006, Bolch et al. 2010) as additional measure. It makes sense that a higher uncertainty is assigned to perennial snow fields. However, a justification for the 30% used would be beneficial. What about the uncertainty of rock glaciers and buried ice? There should at least be some information.

> ***Text was revised to better explain our methods. We do not use the buffer method because a constant buffer because the resulting uncertainty is determined by glacier size (high for small glaciers, low for large glaciers). Image resolution is not an issue because all our imagery was 1 m or better.***

Paragraphs L148ff and L161ff should be moved before the uncertainty paragraph.

> ***Agreed.***

L172ff (Results section): The results section is too thin also for a data paper. More information and analysis (e.g. about topographic variables: min, max, median elevation, hypsometry, slope, aspect, mean elevation vs. distance from the sea, e.g. a figure similar to Fig. 1 but with the mean elevation colour coded would be very valuable) can easily be included and would strongly increase the value of the paper. The other inventory papers in ESSD typically provide information beyond the pure description if the inventory. Moreover, all numbers should be presented along with the uncertainty ranges.

***A paragraph summarizing the topographic variables was added as well as two figures.***

Table 1: Rock glaciers should be an own class.

> ***We cannot include rock glaciers here. We understand the motivation; however those data are the subject of a forthcoming report with its own methodological issues separate from this report. Here we focus solely on glaciers and perennial snowfields.***

I recommend to show the total uncertainty are ranges along with the area (e.g. 10.63 ± 0.61 km²).

> ***Agreed***

The meaning of Max area and Mean area is unclear.

> ***Text revised***

A discussion section is completely missing but would be highly beneficial. Here the authors can discuss methodological challenges and how they overcome along with the literature and in particular compare their inventory with the existing former one(s).

> ***A brief discussion added***

The summary section should be turned into a conclusion section.

> ***Revised***

Do not hesitate to contact me in case you have any question or a comment need clarifications.

Best regards,

Tobias Bolch

References:

Bolch, T., Rohrbach, N., Kutuzov, S., Robson, B.A., Osmonov, A., 2019. Occurrence, evolution and ice content of ice-debris complexes in the Ak-Shiirak, Central Tien Shan revealed by geophysical and remotely-sensed investigations. Earth Surf. Process. Landforms 44, 129–143. https://doi.org/10.1002/esp.4487.

Charbonneau, A.A., Smith, D.J., 2018. An inventory of rock glaciers in the central British Columbia Coast Mountains, Canada, from high resolution Google Earth imagery. Arct. Antarct. Alp. Res. 50, e1489026. https://doi.org/10.1080/15230430.2018.1489026.

Granshaw, F.D., Fountain, A.G., 2006. Glacier change (1958-1998) in the North Cascades National Park Complex, Washington, USA. J. Glaciol. 52, 251–256.

Huss, M., Hock, R., 2018. Global-scale hydrological response to future glacier mass loss. Nature Clim. Change 8, 135–140. https://doi.org/10.1038/s41558-017-0049-x.

RGIK, 2022. Towards standard guidelines for inventorying rock glaciers: baseline concepts (version 4.2.2). IPA Action Group Rock glacier inventories and kinematics, 13 pp. https://bigweb.unifr.ch/Science/Geosciences/Geomorphology/Pub/Website/IPA/CurrentVersion/Current_Baseline_Concepts_Inventorying_Rock_Glaciers.pdf

Janke, J.R., 2007. Colorado Front Range rock glaciers: Distribution and Topographic Characteristics. Arctic, Antarctic, and Alpine Research 39, 74–83.

Janke, J.R., Bolch, T., 2021. Rock glaciers, in: Haritashya, U.K. (Ed.), Treatise on Geomorphology, 2nd Ed. Elsevier. Doi: 10.1016/B978-0-12-818234-5.00187-5

Lukas, S., 2011. Ice-cored moraines, in: Singh, V.P., Singh, P., Haritashya, U.K. (Eds.), Encyclopedia of Snow, Ice and Glaciers. Springer Science+Business Media B.V, Dordrecht, pp. 616–619.

Lukas, S., Nicholson, L. I., Ross, F. H., & Humlum, O. (2005). Formation, meltout processes and landscape alteration of high-Arctic ice-cored moraines. Examples from Nordenskiold Land, Central Spitsbergen. Polar Geography, 29, 157–187.

Mölg, N., Bolch, T., Rastner, P., Strozzi, T., Paul, F., 2018. A consistent glacier inventory for the Karakoram and Pamir regions derived from Landsat data: distribution of debris cover and mapping challenges. Earth Syst. Sci. Data Discuss., 1–44. https://doi.org/10.5194/essd-2018-35.

Østrem, G., Arnold, K., 1970. Ice-cored moraines in southern British Columbia and Alberta, Canada. Geogr. Ann. A 52 A, 120–128.

Paul F. et al. 2013. On the accuracy of glacier outlines derived from remote sensing data. Ann. Glaciol. 54, 171–182.

Pritchard, H.D., 2019. Asia's shrinking glaciers protect large populations from drought stress. Nature 569, 649–654. https://doi.org/10.1038/s41586-019-1240-1.

---

## Author Comment (AC2)

Author Responses in **_bold italics_**

Dear editor, dear authors,

I greatly apologize for the delay in this review, and I hope that its generally positive tone can compensate for the long waiting time. The delay is related to my recent move to another hemisphere and a heavy proposal deadline. Since it is the first time I am reviewing a scientific article in a data format, I was unsure what contents and structure are expected in such publications. Thus, my recommendations focus on the necessary improvements in the clarity and interpretation of the results rather than on requirements for the manuscript structure. Nevertheless, I do agree with Tobias that the format of a technical report is hardly appropriate for a scientific publication in a high-impact journal like the ESSD, with a discussion section painfully missing and the summary section being too brief.

Currently, the manuscript reads almost like a mini article (i.e., a correspondence type), except for its enormous appendix section. Maybe it fits the expected format of ESSD manuscripts, but there are some additions that are necessary to make it easier to follow its arguments and understand the interpretation of the dataset. I hope the recommendations that Tobias and I have provided are useful to achieve such a transformation. Given the above thoughts, I recommend the publication of this manuscript after moderate revisions.

> **_We thank the reviewer for her help in improving our manuscript. We have extensively revised the text to improve clarity and rigor._**

Specific comments:

This article presents an important dataset - a new inventory of glaciers and perennial snowfields for the entire western US that has been long overdue. Most glaciated regions in the world, especially in the "collective West", undergo repeated country- and region-scale inventories of cryospheric landforms on a regular basis, and it is somewhat surprising that such inventories are missing in the western US. I think it would be great to include a paragraph in the introduction section explaining the reasons for why systematic mapping of glacier and snowfield changes has not been done on a regular basis.

> **_The reviewer is correct that an updated inventory of the western US is long overdue and I suppose it is surprising that regular inventory updates are missing. Indeed, the US does a much better job in Antarctica and Greenland. The cause may be due to our science funding structure. The National Science Foundation's Polar Programs is limited, by law, to the Polar Regions. For the western US funding is derived from other broader natural science programs. And I suppose the federal agencies, such as the US Geological Survey, who might normally do such studies are overworked and under-funded. In short, tracking glacier change in the western US is not a priority. For the future, I don't see regular mapping becoming a priority. Regarding the reviewer's recommendation for a paragraph explaining the lack of regular systematic mappings, I don't see how that adds to the report. There are lots of things that should be done that are not._**

I am wondering about the statement in the introduction section (lines 59-61) that "automated schemes are known to misclassify debris-covered ice". I do not agree with this generalization, since most recent

studies have shown that machine learning algorithms can assist the mapping of complicated landforms such as debris-covered and rock glaciers when surface movements (associated with periglacial/glacial processes) are mapped in time and using a combination of satellite products, resulting in a relatively high precision of mapping compared to manual (e.g., Lu et al., 2022; Robson et al., 2020). I think this recent development deserves some space in the introduction and also in the currently missing outlook (i.e., future technological improvements).

> ***Revised***

It is also relevant to mention here the statement from lines 84-85 – if a long-term surface movement could be automatically tracked, one would not need to rely on such superficial criteria as the presence of crevasses. For example, in Norway there are several major ice caps that are nearly completely void of crevasses and thus do not qualify as glaciers according to the established criteria. In contrast, in the Andes, there are glaciers that are surrounded by snow/ice penitents that might appear as crevasses from above but are not part of the glaciers. Such exceptions exist in every glaciated part of the world, albeit with regional differences. Hence, possible future upgrades of the mapping methods need to be discussed in the "future perspectives" to avoid such limiting assumptions.

> ***We agree that surface movement detected from the appearance of crevasses could be improved in the future by automatically tracked movement. Given that the technology is on the cusp of provided that service, we do not feel it needs to be discussed in a 'future perspectives.***

When reading through the text, there are several references to the appendix section that discusses individual cryospheric landforms causing various complications when being mapped or showcasing common unusual behavior. I think it is counterintuitive that in these instances, the article sends its reader to the appendix and its different sections. I believe that many of these belong to the discussion and should be visualized by two additional figures in the main text, including multiple insets corresponding to different issues encountered during mapping and different phenomena observed. These can both feed the non-existing discussion and provide support to the claims in the article that are poorly grounded right now, since the reader must read additional sections from the appendix to visualize and validate them. Furthermore, references to the interesting cases presented in the appendix are not providing a complete picture. I believe they could be strengthened in the main text by adding more examples from the appendix and crafting good visualizations.

> ***The text and figures regarding the difficult cases was moved from the appendix to the main body of the paper.***

The methods section related to the "uncertainty" has two main issues. First, the strategy for estimating uncertainties seems random and non-mathematical (at least in part). Could the authors explain why they decided to adapt such strategies and how they relate to other existing studies with similar objectives (especially, those that are more mathematically grounded)? Ideally, this should be done in the new discussion section. Also, large parts of the uncertainty section read like a discussion to me. I think this is where they should be relocated.

> ***This section was revised to better explain our methods. It should remain in Methods.***

Finally, the results section has two paragraphs of text and is not attached to any discussion. I understand it is a data article but… two paragraphs? Is there nothing more to write about these data in the main text?

*Agreed. Text was revised.*

Technical edits:

Lines 23 – 24: This needs to be expanded. Also, in lines 21 – 34 and in the article, the hazardous nature of glaciers can be included as a motivation (glacial lake outburst floods, glacier collapses and avalanches, etc.) in addition to their function as freshwater resources.

*Considering that this report is about a glacier inventory and not about glacial hazards, we think the brief summary and pointers provided here to the hazard literature is sufficient.*

Line 39: A reference to Andreassen et al. (2022) could be useful here.

*Added*

Lines 107 – 112: This text needs further clarification. It is a bit hard to follow.

*Revised*

Correct grammar/errors in lines 49, 57, 64, 142, and 181.

*Edited all we could find*

References:

Andreassen, L., Nagy, T., Kjøllmoen, B., and Leigh, J. (2022). An inventory of Norway's glaciers and ice-marginal lakes from 2018-19 Sentinel-2 data. Journal of Glaciology, 68(272), 1085-1106, doi: 10.1017/jog.2022.20.

Robson, B.A., Bolch, T., MacDonell, S., Hölbling, D., Rastner, P., and Schaffer, N. (2020). Automated detection of rock glaciers using deep learning and object-based image analysis, Remote Sensing of Environment, 250, 112033, doi: 10.1016/j.rse.2020.112033.

Lu, Y., Zhang, Z., Kong, Y., and Hu, K. (2022). Integration of optical, SAR and DEM data for automated detection of debris-covered glaciers over the western Nyainqentanglha using a random forest classifier, Cold Regions Science and Technology, 193, 103421, doi: 10.1016/j.coldregions.2021.103421.

Citation: https://doi.org/10.5194/essd-2022-369-RC2

---

## Author Comment (AC3)

**Comments to ESSD-2022-369**

**General comments**

The study by Fountain et al. is presenting the results of a new glacier inventory for the contiguous United States /without Alaska) as mapped from manual digitizing of orthorectified digital aerial imagery. Independent of my comments below, I want to congratulate the authors to this long overdue update and acknowledge the great effort that is visible here. I have a number of more general comments to a) the terminology, b) the 'layout of the 'paper' and c) the datasets as well as some more specific ones.

*We thank the reviewer for his attention to detail and thorough review of our manuscript. We have made a number of changes and improvements.*

Starting with a), I am not very happy with mixing glaciers and perennial snowfields. First, snow fields should not be included in a glacier inventory and second, I think the definition applied here for 'perennial snowfields' is ambiguous. Of course, glaciers can be described as perennial snow (L82), as they originate from snow that survives the melting season over several years, but I think this is not the same as 'perennial snow fields' that are just composed of snow and firn and should thus not be included in a glacier inventory. Things get a bit complicated when ice patches (not moving glacier ice) - that might be completely covered by seasonal snow - are to be included but seasonal snow has to be excluded.

*We appreciate the reviewer's consideration regarding glaciers versus perennial snowfields. We feel it important to include the snowfields for their importance to high alpine ecology. In fact, we believe all inventories should include perennial snowfields as they contribute to a comprehensive inventory of perennial ice and snow. We are very clear in the manuscript the separation between the two. We do not mix them. Our criteria to distinguish them are clear and given the time series of imagery available on Google Earth, we are confident about our classification. Also, the other two reviewers did not object to inclusion of snowfields.*

Given that seasonal and perennial snowfields are abundant in this region, that their separation is nearly impossible in many cases, and that the transition of a glacier to an ice patch is gradual, I suggest using a definition that is better suited for this environment. Leigh et al. (2019) have tried to sort this out with a scoring system that can be applied when very high-resolution images are available. I suggest testing it here and re-evaluate the assignment. Currently, a large number of the here assigned 'perennial snowfields' are actually glaciers, e.g. they show bare ice, deformed debris bands, lateral moraines and could be found 'above' units classified as glaciers (and in a few cases its also the other way round, some avalanche deposits in valley floors are classified as glaciers). One example is shown in Figure A10. I think there is no need to assign the class 'Perennial snowfield' to (the two parts of) Freemont glacier. This is still a usual glacier that is actually connected in its lower part (under a thick medial moraine) to the neighbouring Sacagawea Glacier.

*Thank you for making us aware of Leigh et al. (2019), it is a very informative paper, and it has been cited in the revised paper. We were gratified that the paper supported our contention of the detail and accuracy of using high resolution orthorectified*

*imagery for compiling a glacier inventory and the importance of having multiple people viewing each feature.*

*Our region is not particularly unique, most, if not all regions, exhibit abundant seasonal and perennial snowfields. They are inherent in glacier environments. But it is true that the transition from glacier to ice patch/perennial snowfield is gradual and ambiguity exists between very small glaciers and ice patches/snowfields.*

*We were intrigued with Leigh's et al., scoring system to identify glaciers. However, we do not see the utility given the effort. A lengthy and time-consuming reanalysis using an experimental and non-standard procedure of over 2500 features requiring identification and tabulation of 7 criteria for each is beyond the scope of this study. Furthermore, it would result in reclassification of < 5% of the features. We agree, however, that our inventory would be a good test for this procedure in a future study.*

*It is not clear to us what that reclassification adds to our understanding of glacier cover given the extra effort involved. That a tiny glacier is counted as a perennial snowfield or vice versa, might affect the total number of features somewhat but not the total glacier area, which is dominated by the larger glaciers. One justification provided by Leigh et al. (2019) for the scoring system is to identify which small features should be marked for review. Given the pace of climate warming all features should be reviewed with new imagery.*

**We disagree with the comment, "Currently, a large number of the here assigned 'perennial snowfields' are actually glaciers, e.g. they show bare ice, deformed debris bands, lateral moraines and could be found 'above' units classified as glaciers (and in a few cases its also the other way round, some avalanche deposits in valley floors are classified as glaciers)." As we stated in the Methods, we do not distinguish between ice patches and perennial snowfields and refer to all as perennial snowfields. So bare ice can be present. Deformed debris bands may are probably the legacy of past movement, if there are no crevasses ((no crevasses are present)). Isolated lateral and terminal moraines are geological features indicating the past presence of a glacier.**

**Regarding Figure A10, the two Fremont glaciers may be connected to Sacagawea Glacier as you suggest. Or they are perhaps narrowly separated under the medial moraine. But there is no clear evidence either way so we consider them separate features.**

I fully acknowledge the difficulties in performing such an assignment and that in many cases a clear assignment might even be impossible, but currently the number of real glaciers that would be removed from the sample when users exclude the perennial snowfield class is rather high and thus worth revisiting.

*We disagree as mentioned above, that a signification number of real would be removed.*

Moreover, in some regions it seems that perennial snowfields (and even glaciers according to Leigh et al. 2019) have not been mapped.

> ***We disagree with this contention. We utilized three independent inventories, one manual, one based on an automated glacier identification procedure, and a third that merely identified the presence of perennial ice or snow, to compile our inventory and we are confident that it is comprehensive.***

Once this is done, please add example images for the various cases in a multi-panel figure so that readers have a chance to follow the decisions. Maybe also a short note on the class 'Buried Ice': I would not use it. Include it with the glacier outline when it looks like glacier ice under debris-cover and leave it when not. None of the dataset users will do the reassignment, but all are aware and will understand that other interpretations might exist. So please decide as an analyst where to place the glacier outline and leave it with this.

> ***We agree that 'Buried Ice' is a problematic category. The observable evidence certainly suggests the presence of subsurface ice. But there is no suggestion of movement, hence it is what we might call 'dead' ice.***

> ***We received an email from Wilfried Haeberli regarding our report not long after it was first online. In the email he included,***

> ***'Please also note that the term "buried ice" is used in permafrost science as a technical term for surface ice embedded within, or on top of, perennially frozen ground. It could be helpful to clearly state that your inventory uses this term in a different sense: terms like "dead ice", "remains of former glaciers" or the like are more common in corresponding cases.'***

> ***We have modified the report to include that distinction of definition.***

Regarding point b), I also have the impression that the current draft reads more like an internal progress report rather than a paper. There is no problem with being short and to the point, but for example a discussion is completely missing, the information included in the attribute table is not presented, glacier characteristics to be included in a glacier inventory (e.g. minimum, mean, maximum and median elevation, or mean slope and aspect) are neither calculated nor presented and visualized and lots of information is listed in the Appendix without providing a good access (e.g. showing the image footprints) or mapping examples from the various regions with outline overlays to see the decisions taken (also in difficult cases). The text provided in the Appendix comes thus across as rather theoretical descriptors of image conditions and would in this form have a better place in the Supplemental Material. Some of the images in the Appendix, however, should be transferred to the main part and used to illustrate the methods. Please show outline overlays and annotate the images to guide the readers through the decisions made.

> ***Agreed. The text has been modified.***

The digitizing of the new dataset has in general an excellent quality and is a clear improvement over the currently available datasets. However, as mentioned above, the assignment of 'this is a glacier' and 'this a perennial snowfield' seems a bit arbitrary and inconsistent at times. Moreover, some glaciers (and/or perennial snow fields) are seemingly missing.

> ***We have responded to these comments earlier.***

I have compiled a few examples at the end of this review for illustration and suggest revisiting the assignment of all perennial snowfields to really have all glaciers included in the glacier class.

> ***Addressed below, after the image captions***.

Please add an item Class_nr with 1 for glaciers and 2 for PS.

> ***We believe this to be unnecessary, given that databases can be easily searched for words. However, if the editors would like such a change it will be implemented.***

I would also encourage the authors to calculate topographic information for each glacier entity, provide the data in the attribute table of the data file and add some selected illustrations of the dataset characteristics to the text (e.g. maps, scatterplots, bar charts, tables). A shape file providing image footprints (to see which outlines have been derived from which image) would be a most welcome asset.

> ***The intent of this report was to submit updated glacier area to the Glacier Land Ice Monitoring from Space (GLIMS) database for inclusion in the next Randolph Glacier Inventory. Inclusion of the topographic variables were not necessary and we do not believe it important to this report. Neither of the other two reviewers suggested this inclusion. However, if the editors disagree, we will add the topographic data.***

**Specific comments**
L1: conterminous or (first n missing) or contiguous (as in L9)?
> *Changed*

L26/27: What are the criteria to cite these publications? Not all of them are about stream flow.
> ***Yes they do. Dussaillant et al, addresses glacier loss to stream flow in South America, Fountain and Tangborn address streamflow variations caused by glacial runoff, and Moore et al addresses a broader North American perspective on the influence of glaciers on runoff.***

L27/29 (and elsewhere): When referring to contemporary glaciers, I would use glacier instead of glacial (see Cogley et al. 2011)
> *Changed*

L62/64: I suggest not naming it a report when it should be a paper, maybe use this study.
> *Changed*

L67: Please give this part an individual subsection 2.1 (and Uncertainties in L132 to 2.2)
> *Added*

L82: How was the 0.01 km2 size threshold applied before the digitizing?
> *Changed*

L81-87: I suggest applying the classification system suggested by Leigh et al. (2019) to get a better handle on what is a perennial snowfield and what can be named a glacier.

*We disagree as explained earlier*

L90: Shaded reliefs are often ambiguous. I suggest using a flow-direction grid to separate glacier complexes into individual entities.

*In the few cases in which we used shaded relief they were very clear. I can imagine cases where that is not true. Fortunately, there were not present in our terrain.*

L103: In fact, this IS a huge common problem.

*Reworded*

L112 (and elsewhere): Please number all sections in the Supplemental Material and refer here also to this number.

*Changed*

L117: 'once part of the glacier': Couldn't this be checked against the previous inventory?

*The imagery used in this area was quite snowy and we cannot make a definitive interpretation.*

L132: Please give Uncertainties and individual subsection (2.2)

*Done.*

L142: digitizing

*Changed*

L148-170: As mentioned in the general comments, can you please illustrate with a Figure how these datasets (SFI and NLCF) look like and how the merging was done?

*We did not see this mentioned in the general comments. But we did a simple overlay of these data sets, first, the SFI over our initial inventory, then the NLCD over the revised initial inventory. In each chase where the outlines did not overlap, we examined/reexamined each feature.*

L172: Please add an analysis of glacier characteristics as derived from a DEM

*Added*

L190: I suggest moving this table to the Supplemental Material and showing in the text only a figure (bar or pie chart). And remove the 'Buried ice' class. Either include or exclude it.

*We prefer to retain it in the main body of the text.*

L205: I think also Table 2 has a better home in the supplemental Material. This is background information, there is little that can be learned from it.

*Agreed*

L210: Please add a Discussion section

*Done*

L262: Can you explain here why some of these are perennial snowfields while others are glaciers. It is not entirely clear, in particular not for the large ones.

Figures A3, A6 and A9 miss outline overlays. Where is Figure A8?

*Figures A2 and A3 show topographic conditions used to distinguish glaciers/perennial snowfields from rock glaciers. Outlines would interfere with the perspective of field conditions*

*Figure A6 I illustrating what we perceive as buried ice. Outlines would complicate an already complex image, and one that also has a number of lines and arrows.*

*Figure A9, like A2 and A3, shows field conditions. Adding an outline would bias the viewer to our interpretation. The absence of an outline better shows the transitional nature from ice to debris-covered ice, to debris.*

*Miscounted figures there is no Figure A8*

Leigh et al. (2019): https://doi.org/10.1017/jog.2019.50

*Regarding the images and comments below, please remember that our inventory is based on three prior efforts, our own reanalysis, the SFI, and the NLCD. Perhaps the NLCD is the most important one to consider here. That inventory utilized LandSat imagery, 15 m spatial scale every three years from 2001 to 2016 to map surfaces features across the entire USA. Among other landscape features, they identified perennial snow and ice. If we initially missed the features identified in the examples below, and if they were present in the NLCD (or the SFI), we would have examined them for inclusion. That they are not in our inventory indicates to us that they are not perennial. We infer that the comments below are based on one-time imagery. Considering buried ice, we see no signature of buried ice. The suggestion that some features are glaciers not snowfields, we don't see crevasses so they are not moving; the debris bands are relict indicators of past movement. However, if we had locations for the examples below, we would have been happy to examine them further.*

Image examples (World Imagery layer of the ESRI Basemap)

[Figure]

Red: glaciers, yellow/orange: perennial snowfields, green: burried ice, blue: RGI6
x: this is also a glacier. Circle: these should be included, at least as perennial snowfields.
Background image: ESRI

x: glaciers rather than perennial snowfields, Circles: Missing. Background image: ESRI.

[Figure]

I would not include the green part. Background image: ESRI.

[Figure]

This is an avalanche deposit rather than a glacier. Maybe not even a perennial snowfield? Background image: ESRI.

[Figure]

Where is the debris covered part (yellow circle), why is this (x) not a perennial snowfield. Background image: ESRI.

[Figure]

The orange outlines should be glaciers rather than perennial snowfields. x looks like a rock glacier. Background image: ESRI.

[Figure]

The orange outlines (larger polygons) should be glaciers rather than perennial snowfields. Background image: Copernicus Sentinel-2 2020

[Figure]

The orange outlines should be glaciers rather than perennial snowfields. Background image: ESRI.